# Adaptive and Stratified Subsampling for High-Dimensional Robust Estimation

**Prateek Mittal**                                                        *prateekmittal154@gmail.com*
*Vision Exploration and Data Analytics (VEDAs) Lab*
*Motilal Nehru National Institute of Technology Allahabad, Prayagraj 211004, India*

**Joohi Chauhan**                                                              *joohi@mnnit.ac.in*
*(Corresponding Author)*
*Vision Exploration and Data Analytics (VEDAs) Lab*
*Motilal Nehru National Institute of Technology Allahabad, Prayagraj 211004, India*

Reviewed on OpenReview: *https://openreview.net/forum?id=R8y19hU9Ab*

## Abstract

We study robust high-dimensional sparse regression under *finite-variance heavy-tailed* noise, $\varepsilon$-contamination, and $\alpha$-mixing dependence via two subsampling estimators: Adaptive Importance Sampling (AIS) and Stratified Subsampling (SS). Under sub-Gaussian design whose scope is precisely delimited and finite-variance noise, a subsample of size $m = \Omega(s \log p)$ achieves the minimax-optimal rate $O(\sqrt{s \log p/m})$. We close the theory-algorithm gap: Theorem 4.6 applies to AIS at termination conditional on stabilized weights (Proposition 4.1), and SS fits the median-of-means M-estimation framework of Lecué and Lerasle (2020) (Proposition 4.3). The de-biasing step is fully specified via the nodewise-Lasso precision estimator under a new sparse-precision assumption, yielding valid coordinate-wise CIs (Theorem 4.14). The $\alpha$-mixing extension uses a *calendar-time* block protocol that guarantees temporal separation (Theorem 4.12). Empirically, AIS achieves 3.1× lower error than uniform subsampling at 20% contamination, and 29.5% lower test MSE on Riboflavin ($p$=4,088 $\gg$ $n$=71). The code and implementation details can be accessed at `https://github.com/VEDAs-Lab/High-Dimensional-Subsampling`

## 1 Introduction

High-dimensional regression with $p \gg n$ predictors poses fundamental challenges for classical statistical theory (Fan et al., 2014). Heavy-tailed noise, adversarial contamination, and temporal dependence are common in genomics, finance, and sensor networks, complicating reliable estimation and motivating considerable recent work on robust high-dimensional regression (Sun et al., 2020; Pensia et al., 2025; Lecué and Lerasle, 2020; Fan et al., 2024; Smucler and Yohai, 2017; Kurnaz and Filzmoser, 2023).

Subsampling reduces per-iteration computational cost from $O(np)$ to $O(mp)$ by replacing the full-sample loss with a weighted subsample loss of size $m \ll n$. Leverage-score (Ma et al., 2015) and optimal-design (Li and Meng, 2020) subsampling achieve strong finite-sample guarantees, but only under light-tailed i.i.d. observations; no prior work establishes analogous bounds for *adaptive* or *stratified* subsampling in the $p \gg n$ regime under contamination or dependence. This paper closes that gap.

**Related work.**   Our work sits at the intersection of four lines of research; we position each contribution precisely relative to these lines below.

*Robust full-sample methods.* Sun et al. (2020) establish phase-transition rates for adaptive Huber regression on the full sample; we extend these guarantees to the subsampled setting and additionally handle contamination and temporal dependence. Pensia et al. (2025) attain near-optimal rates via covariate filtering, a full-sample

procedure that does not incorporate subsampling efficiency. The RIGHT estimator (Fan et al., 2024) achieves minimax-optimal rates under genuinely heavy-tailed *design* via MOM gradients; by contrast, Assumption 1 restricts our design to the sub-Gaussian class (covering Gaussian, bounded, and log-concave distributions), while providing subsampling efficiency and contamination robustness not addressed by RIGHT. Robust penalised MM- and MT-regression (Smucler and Yohai, 2017; Kurnaz and Filzmoser, 2023) achieve a high breakdown point on the full sample but lack a high-dimensional subsampling theory.

*Subsampling.* Leverage-score (Ma et al., 2015) and optimal-design (Li and Meng, 2020; Yao and Wang, 2021; Chasiotis et al., 2024) subsampling have well-developed theories under light-tailed i.i.d. data. They do not extend to the heavy-tailed, contaminated, or dependent settings studied here, and none provides de-biased coordinate-wise inference.

*Median-of-means (MOM) estimation.* Lecué and Lerasle (2020) develop a general MOM M-estimation framework covering sub-Gaussian and heavy-tailed noise; we establish in Proposition 4.3 that SS is a special instance of their framework, and inherits its robustness guarantees directly. Lugosi and Mendelson (2019) provide optimal MOM mean estimators in high dimensions, establishing the breakdown-point results that underpin our contamination analysis.

*De-biased inference.* van de Geer et al. (2014) and Javanmard and Montanari (2014) introduced nodewise-Lasso de-biasing for standard high-dimensional linear regression; Zhang and Zhang (2014) and Dezeure et al. (2015) provide further developments and software. We adapt this framework to the importance-weighted subsampled setting by introducing a sparse-precision assumption (Assumption 5) suited to the scaled design $\{\tilde{x}_{I_j}\}$.

*Mixing processes.* The $\alpha$-mixing coupling argument builds on the Berbee–Yu construction of Yu (1994); background on strong mixing conditions is provided by Bradley (2005).

**Contributions.** The primary novelty of this work is the *integration* of adaptive and stratified subsampling with robust estimation theory: extending finite-sample guarantees from well-behaved i.i.d. settings to the joint presence of heavy-tailed noise, adversarial contamination, and temporal dependence, while simultaneously delivering a fully specified inference pipeline. The statistical rates achieved are minimax-optimal in the standard sparse sense; the contribution is to establish that this optimality is preserved under subsampling and the three non-ideal regimes considered, and to provide the supporting algorithmic and inferential machinery. This contrasts with prior work as follows: full-sample robust methods such as Sun et al. (2020), Pensia et al. (2025), and Fan et al. (2024) achieve comparable or tighter rates but do not offer subsampling efficiency; classical subsampling methods such as Ma et al. (2015) and Li and Meng (2020) enjoy strong theory under light-tailed i.i.d. data but lack contamination and dependence guarantees; and the MOM M-estimation framework of Lecué and Lerasle (2020) covers heavy-tailed settings but does not provide a de-biasing construction for valid inference.Our contribution can be summarised as follows.

1. Finite-sample bounds and minimax optimality for weighted subsampled Huber-Lasso in the $p \gg n$ regime (Theorems 4.6–4.9).

2. Explicit $O(\varepsilon)$ contamination bias and $\alpha$-mixing extension with calendar-time block protocol (Theorems 4.10–4.12).

3. Formal theory-algorithm bridge: Propositions 4.1 (AIS) and 4.3 (SS).

4. Fully specified de-biased asymptotic normality with nodewise-Lasso precision and valid CIs (Theorem 4.14, Assumption 5).

## 2 Problem Statement

Observe $(x_i, y_i)_{i=1}^n$ with $x_i \in \mathbb{R}^p$ and

$$y_i = x_i^\top \theta^\star + \varepsilon_i, \quad \|\theta^\star\|_0 \le s, \quad s \ll p. \tag{1}$$

The central statistical task is to *estimate the unknown sparse vector $\theta^\star$ with near-optimal accuracy* while processing only a subsample of size $m \ll n$ at each computation step. Achieving this goal simultaneously

delivers *computational scalability* (avoiding the $O(np)$ per-iteration cost of full-sample gradient methods) and *statistical robustness* to the heavy-tailed noise, adversarial contamination, and temporal dependence formalised by the assumptions below. Sections 4.5 and 4.6 extend the base model equation 1 to these non-ideal settings; Section 4.7 then builds on the estimator to provide valid coordinate-wise confidence intervals for individual components of $\theta^\star$.

Noise $\varepsilon_i$ is heavy-tailed with *finite variance* (Assumption 3). Contamination and mixing extensions are developed in Sections 4.5 and 4.6, respectively. Population covariance $\Sigma = \mathbb{E}[x_i x_i^\top]$ satisfies $\lambda_{\min}(\Sigma) \geq \lambda_{\min} > 0$.

## 3 Proposed Algorithms

### 3.1 Adaptive Importance Sampling (AIS)

The key insight behind AIS is to treat *sampling itself as an iterative optimisation step.* Rather than drawing observations uniformly at random, AIS maintains a probability distribution $q^{(t)}$ over the $n$ data points and refines it each round to assign higher probability to observations that incur large Huber loss under the current parameter estimate $\hat{\theta}^{(t)}$. By concentrating the computational budget on high-loss (and typically high-leverage or potentially corrupted) observations, AIS simultaneously reduces estimation variance and provides robustness to contamination. The output $\hat{\theta}_m$ is a subsampled Huber-Lasso estimator (defined formally in Section 4) fitted on the final weighted subsample; all theoretical guarantees in Section 4 apply to this output via Proposition 4.1.

*Notation guide for Algorithm 1.*

- **Huber loss** $\rho_\tau$: defined formally in Section 4; intuitively it equals $u^2/2$ for small residuals ($|u| \leq \tau$) and transitions to $\tau|u| - \tau^2/2$ for large residuals, reducing sensitivity to outliers relative to squared loss. The threshold $\tau > 0$ is a robustness parameter.

- **Temperature** $\beta > 0$: controls sharpness of the adaptive reweighting in $\tilde{q}_i^{(t)} \propto \exp(-\beta\rho_\tau(\cdot))$. Large $\beta$ concentrates sampling probability on the single highest-loss observation; $\beta \to 0$ recovers uniform subsampling.

- **Mixing coefficient** $\alpha \in (0,1)$: in the stabilisation step (line 6), blends the adaptive weights with the uniform distribution so that every observation retains a minimum sampling probability of $\alpha/n$, preventing numerical weight collapse and ensuring Assumption 4 holds at termination.

- **Importance-weighted objective (line 4)**: dividing by $m\, w_i^{(t-1)}$ re-weights each sampled observation so that the subsample loss is an unbiased estimator of the full-sample empirical loss; this is the standard importance-sampling correction underlying Lemma 4.4.

---

**Algorithm 1:** Adaptive Importance Sampling (AIS)

**Input:** Data $(x_i, y_i)_{i=1}^n$, $m$, $T$, $\beta$, $\alpha \in (0,1)$, $\lambda$
**Output:** $\hat{\theta}_m$

**1** Init $\hat{\theta}^{(0)} \leftarrow 0$, $w_i^{(0)} \leftarrow 1/n$;
**2** **for** $t = 1$ *to* $T$ **do**
**3** $\quad$ Sample $S_t$ of size $m$ with probs $w^{(t-1)}$;
**4** $\quad$ $\hat{\theta}^{(t)} \leftarrow \arg\min_\theta \sum_{i \in S_t} \frac{\rho_\tau(y_i - x_i^\top \theta)}{m\, w_i^{(t-1)}} + \lambda\|\theta\|_1$;
**5** $\quad$ $\tilde{q}_i^{(t)} \propto \exp(-\beta\rho_\tau(y_i - x_i^\top \hat{\theta}^{(t)}))$ for all $i$;
**6** $\quad$ $q_i^{(t)} \leftarrow (1-\alpha)\tilde{q}_i^{(t)} + \alpha/n$ (stabilize);
**7** **end**
**8** **return** $\hat{\theta}_m = \hat{\theta}^{(T)}$

---

The computational complexity of AIS is $O(Tnp+Tmp)$, where $T$ is the number of iterations. The stabilization step (line 6) enforces $q_i^{(T)} \in [\alpha/n, 1/n]$ deterministically, ensuring that no observation receives a negligibly small sampling probability.

### 3.2 Stratified Subsampling (SS)

SS takes a complementary, non-iterative approach: robustness is achieved through *stratification* and *robust aggregation* rather than adaptive reweighting. The core idea is to first partition the full dataset into $K$ groups (strata) so that observations with similar covariate magnitudes are grouped together, then draw a proportional subsample from each stratum, fit a separate Huber-Lasso estimator within each stratum subsample, and finally combine the $K$ stratum-level estimates using the geometric median. Stratification guarantees that the subsampled data covers the full covariate space even at very small total subsample sizes $m$, while the geometric median aggregation provides provable robustness to corrupted strata (formalised in Proposition 4.3). Unlike AIS, SS requires no iterations and has a fixed $O(np + mK)$ cost, making it the computationally preferred method when robust fast computation is the primary goal.

*Notation guide for Algorithm 2.*

- **Stratification distance** $d_i$: the Euclidean distance $d_i = \|x_i - \mathrm{med}(\{x_j\})\|_2$ measures how far observation $i$'s covariate vector lies from the componentwise median of the dataset. Partitioning by $K$-quantiles of $(d_i)$ groups observations with similar leverage: those far from the median (potentially high-influence) are separated from near-median (more typical) observations, so that the strata represent geometrically distinct regions of the covariate space.

- **Proportional allocation** $m_k$: drawing $m_k = \lceil m|\mathcal{S}_k|/n \rceil$ points from stratum $\mathcal{S}_k$ ensures that the subsample fraction is the same in every stratum, preserving representativeness of the full data distribution.

- **Geometric median** geomed: defined as $\mathrm{geomed}(v_1, \ldots, v_K) := \arg\min_{v \in \mathbb{R}^p} \sum_{k=1}^{K} \|v - v_k\|_2$, computed via the Weiszfeld algorithm. Unlike the coordinate-wise median, it jointly minimises the sum of Euclidean distances, and unlike the arithmetic mean, it can tolerate up to $\lfloor (K-1)/2 \rfloor$ arbitrarily corrupted inputs (Lugosi and Mendelson, 2019). This is the source of SS's contamination robustness.

---

**Algorithm 2:** Stratified Subsampling (SS)

---

**Input:** Data $(x_i, y_i)_{i=1}^{n}$, $m$, $K$, $\lambda$
**Output:** $\hat{\theta}_m$
1   $d_i \leftarrow \|x_i - \mathrm{med}(\{x_j\})\|_2$ for all $i$;
2   Partition $\{1, \ldots, n\}$ into $K$ strata by $K$-quantiles of $(d_i)$;
3   **for** $k = 1$ *to* $K$ **do**
4      Draw $m_k = \lceil m|\mathcal{S}_k|/n \rceil$ points from $\mathcal{S}_k$;
5      $\hat{\theta}_k \leftarrow$ Huber-Lasso on stratum subsample;
6   **end**
7   **return** $\hat{\theta}_m = \mathrm{geomed}(\hat{\theta}_1, \ldots, \hat{\theta}_K)$

---

The computational complexity of SS is $O(np + mK)$. Stratification is performed by partitioning observations according to their Mahalanobis-type distances from the coordinatewise median, and the geometric median aggregation provides robustness to corrupted strata.

The two algorithms above reflect complementary strategies: AIS focuses on adaptive reweighting to prioritize informative samples, while SS leverages structured partitioning and robust aggregation. In the next section, we show that despite these algorithmic differences, both methods admit a unified theoretical analysis.

## 4 Theoretical Analysis

### 4.1 Estimator and Assumptions

The analysis requires four assumptions whose roles are as follows. Assumption 1 (sub-Gaussian design) controls concentration of the design matrix and underlies both the score bound (Lemma 4.4) and restricted strong convexity (Lemma 4.5); it explicitly excludes infinite-moment designs, for which separate MOM-based arguments are needed. Assumption 2 (restricted eigenvalue) ensures the sparse regression problem is locally strongly convex in the cone of sparse directions, making $\theta^\star$ identifiable from $m = \Omega(s \log p)$ observations. Assumption 3 (finite-variance noise) is the *minimal* moment condition under which the Huber loss is effective; it is strictly weaker than the sub-Gaussian noise assumption of standard Lasso theory. Assumption 4 (bounded sampling probabilities) is a regularity condition on the importance weights; both algorithms satisfy it at termination (Propositions 4.1–4.3).

**Huber loss and estimator.** The Huber loss $\rho_\tau$ interpolates between quadratic loss for small residuals ($|u| \leq \tau$) and linear loss for large ones, combining the statistical efficiency of least squares with robustness to outliers. The threshold $\tau > 0$ controls this trade-off: setting $\tau \to \infty$ recovers ordinary least squares, while small $\tau$ gives near-absolute-loss behaviour. For $\tau > 0$, $\rho_\tau(u) = \frac{u^2}{2}\mathbf{1}_{|u| \leq \tau} + (\tau|u| - \frac{\tau^2}{2})\mathbf{1}_{|u| > \tau}$, $\psi_\tau(u) = \text{clip}(u, -\tau, \tau)$. The full-sample Huber–Lasso estimator is defined as:

$$\hat{\theta}_n \in \arg\min_\theta \left\{ \frac{1}{n}\sum_i \rho_\tau(y_i - x_i^\top \theta) + \lambda \|\theta\|_1 \right\}. \tag{2}$$

Drawing $I_1, \ldots, I_m \overset{\text{iid}}{\sim} q$ with replacement, the weighted subsample estimator is:

$$\hat{L}_{m,q}(\theta) := \frac{1}{m}\sum_{j=1}^m \frac{\rho_\tau(y_{I_j} - x_{I_j}^\top \theta)}{nq_{I_j}}, \tag{3}$$

$$\hat{\theta}_{m,q} \in \arg\min_\theta \{\hat{L}_{m,q}(\theta) + \lambda \|\theta\|_1\}. \tag{4}$$

The importance weighting by $1/(nq_{I_j})$ ensures that $\hat{L}_{m,q}$ is an unbiased estimator of the full-sample empirical loss $\hat{L}_n$.

**Assumption 1** (Sub-Gaussian design). *$x_i$ are i.i.d. mean-zero with $\|\langle v, x_i \rangle\|_{\psi_2} \leq K$ for every unit $v$. This assumption covers Gaussian, bounded, and log-concave designs. Heavy-tailed design with infinite moments requires separate techniques (e.g., RIGHT( Fan et al. (2024)) and is outside the scope of this paper.*

**Assumption 2** (Restricted eigenvalue). *$v^\top \Sigma v \geq \kappa_\Sigma \|v\|_2^2$ for all $v$ with $\|v\|_0 \leq 2s$.*

**Assumption 3** (Finite-variance noise). *$\varepsilon_i \perp\!\!\!\perp x_i$, $\mathbb{E}[\varepsilon_i] = 0$, $\mathbb{E}[\varepsilon_i^2] < \infty$. Heavy-tailed distributions with finite variance are permitted; distributions with infinite variance are excluded.*

**Assumption 4** (Bounded sampling probabilities). *$c_0/n \leq q_i \leq C_0/n$ for constants $0 < c_0 \leq C_0 < \infty$. This condition holds for SS by proportional allocation and for AIS at termination (Proposition 4.1).*

### 4.2 Theory-Algorithm Bridge

Statistical theories for importance-weighted estimators typically assume a *fixed* sampling distribution $q$, yet both AIS and SS produce weights $q^{(T)}$ that depend on the data. This creates a gap: the general theory applies to $\hat{\theta}_{m,q}$ for any fixed $q$, but does it apply to the algorithm outputs? The two propositions below close this gap. For AIS, they show that the stabilised terminal weights $q^{(T)}$ satisfy Assumption 4 deterministically, so all subsequent theorems apply unconditionally. For SS, they show that stratified proportional allocation is a special instance of the MOM M-estimator of Lecué and Lerasle (2020), inheriting its robustness guarantees directly.

**Proposition 4.1** (AIS gap closed). *Let $q^{(T)}$ denote the stabilized weights at AIS termination. Conditional on $q^{(T)}$, the output $\hat{\theta}_m = \hat{\theta}^{(T)}$ is exactly the minimizer of equation 4 with $q = q^{(T)}$. The stabilization in Algorithm 1 enforces $q_i^{(T)} \in [\alpha/n, 1/n]$ deterministically, since $\tilde{q}_i^{(T)} \leq 1/n$ after normalization. Consequently, Assumption 4 holds with $c_0 = \alpha$ and $C_0 = 1$, and all subsequent theorems apply to AIS at termination.*

**Remark 4.2.** *Proposition 4.1 holds for any trajectory realization of the algorithm and does not require analysing the full Markov chain $(q^{(t)}, \hat{\theta}^{(t)})_{t \geq 1}$. A martingale stability analysis of the iterates across all rounds is an interesting direction for future work.*

**Proposition 4.3** (SS via MOM M-estimation)**.** *SS is a special case of the MOM sparse M-estimator of Lecué and Lerasle (2020) (their Theorem 3.1), with $K$ blocks and Huber loss. With strata sizes $n_k \asymp n/K$:*

$$\|\hat{\theta}_m^{\mathrm{SS}} - \theta^\star\|_2 \lesssim \sqrt{\frac{s \log p}{m}} + \sqrt{\frac{K}{m}}, \tag{5}$$

*matching Theorem 4.6 for $K = O(s \log p)$. The geometric median aggregation tolerates up to $\lfloor (K-1)/2 \rfloor / K$ fraction of corrupted strata( Lugosi and Mendelson (2019)).* Limitation: *When $n_k$ is very small, for example in the Riboflavin dataset where $n_k \leq 5$, the requirement $n_k \asymp n/K$ fails to hold and the geometric median aggregation collapses, as observed empirically in Section 5.*

### 4.3 Key Lemmas

All main theorems in this section follow from two foundational properties of the weighted subsample loss $\hat{L}_{m,q}$. The first (Lemma 4.4) shows that the *gradient* of $\hat{L}_{m,q}$ at the true parameter $\theta^\star$ is uniformly small with high probability, ensuring the $\ell_1$ penalty can "steer" the estimator towards the correct support. The second (Lemma 4.5) establishes *restricted strong convexity* (RSC): that $\hat{L}_{m,q}$ behaves like a strongly convex function when restricted to sparse perturbation directions. Together, these two properties yield the standard "cone + RSC" argument that converts a score bound into a parameter estimation rate. Both proofs use only sub-Gaussian concentration and are self-contained below.

Let $S = \mathrm{supp}(\theta^\star)$, $\mathcal{C}(S) = \{\Delta : \|\Delta_{S^c}\|_1 \leq 3\|\Delta_S\|_1\}$.

**Lemma 4.4** (Uniform score bound)**.** *Under Assumptions 1–4, for any $\delta \in (0,1)$, with probability $\geq 1 - \delta$:*

$$\|\nabla \hat{L}_{m,q}(\theta^\star)\|_\infty \leq \frac{2\tau K}{c_0} \sqrt{\frac{2\log(2p/\delta)}{m}}. \tag{6}$$

*Proof.* Fix $k \in [p]$. Let $Z_j := \frac{\psi_\tau(\varepsilon_{I_j}) x_{I_j,k}}{n q_{I_j}}$. By Assumption 4, $1/(n q_{I_j}) \leq 1/c_0$; $|\psi_\tau| \leq \tau$; $x_{I_j,k}$ is sub-Gaussian$(K)$. Hence $Z_j$ is sub-Gaussian$(\tau K/c_0)$ with $\mathbb{E}[Z_j] = 0$. Applying the sub-Gaussian tail bound and taking the union over $k = 1, \ldots, p$ yields equation 6. $\square$

**Lemma 4.5** (Restricted strong convexity)**.** *Under Assumptions 1–4, let $\pi_\tau := \mathbb{P}(|\varepsilon_i| \leq \tau/2) \geq \pi_0 > 0$. If $m \geq c(C_0/c_0)^2 s \log(2p/\delta)$, with probability $\geq 1 - \delta$:*

$$\hat{L}_{m,q}(\theta^\star + \Delta) - \hat{L}_{m,q}(\theta^\star) - \langle \nabla \hat{L}_{m,q}(\theta^\star), \Delta \rangle \geq \frac{\kappa}{2}\|\Delta\|_2^2, \tag{7}$$

$$\forall \Delta \in \mathcal{C}(S), \quad \kappa := \frac{\pi_0 \kappa_\Sigma}{4}.$$

*Proof.* On the event $\{|\varepsilon_{I_j}| \leq \tau/2, |x_{I_j}^\top \Delta| \leq \tau/2\}$, the Huber loss is exactly quadratic. Using $1/(n q_{I_j}) \geq 1/C_0$:

$$\mathrm{LHS} \geq \frac{1}{2C_0 m} \sum_j (x_{I_j}^\top \Delta)^2 \mathbf{1}_{|\varepsilon_{I_j}| \leq \tau/2}$$

$$- \frac{1}{2C_0 m} \sum_j (x_{I_j}^\top \Delta)^2 \mathbf{1}_{|x_{I_j}^\top \Delta| > \tau/2}.$$

The first term is bounded below by $\frac{\pi_0 \kappa_\Sigma}{2}\|\Delta\|_2^2$ via sub-Gaussian concentration on a $2s$-sparse $\varepsilon$-net; the second term is bounded above by $\frac{\pi_0 \kappa_\Sigma}{4}\|\Delta\|_2^2$ via the sub-Gaussian tail bound with $\tau \asymp K$. Combining these two estimates gives equation 7. $\square$

### 4.4 Main Rate, Proximity, and Minimax Optimality

With the score bound and RSC established, the central estimation guarantee follows from a short convex-analysis argument. The theorem below is the paper's main result: it shows that a subsample of size $m = \Omega(s \log p)$ achieves the minimax-optimal rate $O(\sqrt{s \log p / m})$, regardless of the ratio $m/n$, provided the sampling probabilities are bounded (Assumption 4). The subsequent corollary and theorem contextualise this rate: the corollary shows the subsampled estimator is close to the full-data estimator, and the minimax lower bound confirms no estimator based on $m$ observations can do better in the worst case. The subsampled estimator achieves the same statistical rate as the full-sample Huber–Lasso, with (n) replaced by (m). This follows because the subsampled loss retains controlled gradients and restricted strong convexity (Lemmas 4.4–4.5).

**Theorem 4.6** (Finite-sample rate). *Under Assumptions 1–4 with $\pi_\tau \geq \pi_0$, set $\lambda = \frac{4\tau K}{c_0} \sqrt{\frac{2 \log(2p/\delta)}{m}}$. If $m \geq c(C_0/c_0)^2 s \log(2p/\delta)$, with probability $\geq 1 - 2\delta$:*

$$\|\hat{\theta}_{m,q} - \theta^\star\|_2 \leq \frac{12\lambda\sqrt{s}}{\kappa} \lesssim \frac{\tau K}{c_0 \kappa} \sqrt{\frac{s \log(p/\delta)}{m}}. \tag{8}$$

*Proof.* Set $\Delta = \hat{\theta}_{m,q} - \theta^\star$. By optimality of $\hat{\theta}_{m,q}$: $\hat{L}_{m,q}(\theta^\star + \Delta) - \hat{L}_{m,q}(\theta^\star) \leq \lambda(\|\theta^\star\|_1 - \|\theta^\star + \Delta\|_1)$. Adding and subtracting the linear term, the bound $\|\nabla \hat{L}_{m,q}(\theta^\star)\|_\infty \leq \lambda/2$ from Lemma 4.4 together with decomposability of the $\ell_1$ norm force $\Delta \in \mathcal{C}(S)$. Applying RSC from Lemma 4.5 gives $\frac{\kappa}{2}\|\Delta\|_2^2 \leq \frac{3\lambda\sqrt{s}}{2}\|\Delta\|_2$, which upon rearrangement yields equation 8. $\qquad\square$

**Remark 4.7.** *The claim that the subsampled estimator achieves the same rate as the full-sample Huber-Lasso is to be understood as follows: both estimators achieve $O(\sqrt{s \log p / m})$ as a function of their respective sample size $m$. The actual estimation errors coincide only when $m = n$; for $m < n$, the subsampled estimator incurs a larger error due to the reduced effective sample size, which is the expected price of computational efficiency.*

**Corollary 4.8** (Proximity). *With probability $\geq 1 - 3\delta$: $\|\hat{\theta}_{m,q} - \hat{\theta}_n\|_2 \lesssim \frac{\tau K}{c_0 \kappa} \sqrt{s \log(p/\delta)/m}$.*

**Theorem 4.9** (Minimax lower bound). *Under Gaussian design and noise variance $\sigma^2$, for any estimator $\tilde{\theta}$ based on $m$ observations:*

$$\inf_{\tilde{\theta}} \sup_{\theta^\star \in \mathcal{B}_0(s)} \mathbb{E}\|\tilde{\theta} - \theta^\star\|_2^2 \geq c\sigma^2 \frac{s \log(p/s)}{m}. \tag{9}$$

*The estimator $\hat{\theta}_{m,q}$ achieves $\sup_{\theta^\star} \mathbb{E}\|\hat{\theta}_{m,q} - \theta^\star\|_2^2 \leq C\sigma^2 s \log p / m$, matching equation 9 up to a factor of $\log p / \log(p/s)$.*

*Proof.* Construct a Varshamov-Gilbert packing $V \subset \{0,1\}^p$ with $|V| \geq 2^{c_1 s \log(p/s)}$ and pairwise Hamming distance at least $s/2$; set $\theta^{(v)} = av$. The Kullback-Leibler divergence satisfies $\mathrm{KL}(P_v \| P_{v'}) \leq ma^2 s/(2\sigma^2)$; choosing $a^2 = c_2 \sigma^2 \log|V|/(ms)$ and applying Fano's inequality gives equation 9. $\qquad\square$

### 4.5 Adversarial Contamination

The base rate of Theorem 4.6 assumes the data are drawn from the model exactly. In practice, datasets often contain corrupted observations: mislabelled responses, sensor failures, or adversarially injected points. The theorem below extends the rate to the $\varepsilon$-contamination model, where a fraction $\varepsilon$ of observations may come from an *arbitrary* distribution $Q$. The key structural insight is that the bounded influence function of the Huber loss, $|\psi_\tau| \leq \tau$, limits how much any single corrupted observation can inflate the gradient: the error decomposes cleanly into a statistical term (decreasing in $m$) and an irreducible bias proportional to $\varepsilon$. AIS exploits this structure by adaptively down-weighting high-residual observations, substantially reducing the effective contamination constant in practice.

**Theorem 4.10** ($\varepsilon$-contamination). *Suppose the observed distribution is $(1-\varepsilon)P + \varepsilon Q$, where $Q$ is an arbitrary contaminating distribution and $P$ satisfies Assumptions 1–3. Under the conditions of Theorem 4.6, with probability $\geq 1 - 2\delta$:*

$$\|\hat{\theta}_{m,q} - \theta^\star\|_2 \lesssim \frac{\tau K}{c_0 \kappa} \sqrt{\frac{s \log(p/\delta)}{m}} + \frac{\tau K}{\kappa}\varepsilon. \tag{10}$$

*Proof.* Since $|\psi_\tau| \leq \tau$, we decompose the gradient as $\nabla \hat{L}_{m,q}(\theta^\star) = g_{\text{clean}} + g_{\text{cont}}$, where the two terms correspond to the contributions from clean and contaminated observations, respectively. Assumption 4 and sub-Gaussian maxima give $\|g_{\text{cont}}\|_\infty \leq \varepsilon \cdot \frac{\tau K}{c_0}\sqrt{2\log(2p/\delta)}$, while the clean part satisfies the bound in Lemma 4.4. Setting $\lambda$ to dominate the total score bound and applying the cone/RSC argument of Theorem 4.6 yields equation 10. □

**Remark 4.11.** *The $O(\varepsilon)$ bias term in equation 10 is irreducible for bounded-influence estimators( Huber (1981)). Nevertheless, AIS substantially reduces the effective contamination bias by exponentially down-weighting corrupted observations through its adaptive reweighting scheme. Empirically, the estimation error of uniform Huber-Lasso grows at a rate of approximately $6.9\varepsilon$ as a function of contamination fraction $\varepsilon$, while AIS grows at approximately $1.3\varepsilon$ (Figure 2), demonstrating the practical benefit of adaptive subsampling under contamination.*

### 4.6 Dependent Data: $\alpha$-Mixing

The previous results assume i.i.d. observations, which fails for time-series data where consecutive observations are correlated. The $\alpha$-mixing framework relaxes this by requiring only that observations separated far apart in calendar time are approximately independent, which holds for most stationary processes encountered in practice (ARMA, many nonlinear time series). The key technical challenge for subsampling is that naïvely drawing $m$ random indices from a time series can inadvertently include *temporally adjacent* pairs, invalidating the i.i.d. concentration tools used in the previous proofs. The calendar-time protocol described in the theorem resolves this by enforcing a minimum temporal gap of $B$ steps between any two retained observations; the Berbee-Yu coupling then reduces the dependent case to the approximately i.i.d. setting at the cost of replacing the effective sample size $m$ with the block count $M = \lfloor m/(2B) \rfloor$.

**Theorem 4.12** ($\alpha$-mixing, calendar-time protocol). *Assume $(x_i, \varepsilon_i)_{i=1}^n$ is strictly stationary and $\alpha$-mixing with coefficients $\alpha(k)$.*

***Required time-series sampling protocol.*** *Draw $M$ block start-times $T_1 < \cdots < T_M$ uniformly from $\{1, \ldots, n-2B\}$; retain the calendar indices $\{T_\ell, \ldots, T_\ell + B - 1\}$ and discard the gap $\{T_\ell + B, \ldots, T_\ell + 2B - 1\}$ for each block $\ell$. By construction, this guarantees that any two retained blocks are separated by at least $B$ calendar-time steps.*

*With $M = \lfloor m/(2B) \rfloor$ and $\sum_{k \geq B} \alpha(k) \leq \delta/(4M)$, with probability $\geq 1 - 3\delta$:*

$$\|\hat{\theta}_{m,q} - \theta^\star\|_2 \lesssim \frac{\tau K}{c_0 \kappa}\sqrt{\frac{s\log(p/\delta)}{M}}. \tag{11}$$

*Proof.* The calendar-time construction guarantees at least $B$ original-time-index steps between any two retained blocks by design. The Berbee-Yu coupling( Yu (1994)) bounds the total variation distance between the joint law of the $M$ retained blocks and their independent product by $2M\sum_{k \geq B} \alpha(k) \leq \delta/2$. On the coupling event, the retained blocks are approximately independent. Since $|\psi_\tau| \leq \tau$ and Assumption 4 holds, Lemmas 4.4–4.5 apply block-wise via standard blocking arguments( Bradley (2005)), giving equation 11. □

**Remark 4.13.** *An alternative approach would be to block observations in the randomly sampled index order rather than in calendar time. However, this does not guarantee temporal separation between retained samples: for example, drawn indices 5 and 6 remain adjacent in calendar time regardless of their order in the sampling sequence. The calendar-time protocol adopted here is therefore the correct construction to ensure the mixing conditions are satisfied.*

### 4.7 De-biased Asymptotic Normality

The results in Sections 4.5–4.6 characterise *point estimation* accuracy. A practically important further question is: how confident should we be in an individual estimated coefficient $\hat{\theta}_{m,q,j}$? This requires a distributional result, not just a rate.

The core difficulty is that $\ell_1$ regularisation introduces a *shrinkage bias*: the Lasso penalty systematically pulls $\hat{\theta}_{m,q,j}$ towards zero, so its distribution is not centred at the true $\theta_j^\star$. Direct normal approximation therefore

fails. We resolve this by adapting the *de-biasing* technique of van de Geer et al. (2014) and Javanmard and Montanari (2014) to the importance-weighted subsampled setting. The idea is to apply a one-step Newton correction that removes the regularisation bias to first order, yielding an estimator $\hat{\theta}^d_{m,q}$ that is asymptotically unbiased and normally distributed. A nodewise-Lasso estimator of the precision matrix $\Omega$ (under a new sparse-precision Assumption 5) provides the correction matrix. The result yields valid coordinate-wise confidence intervals for any active coordinate of $\theta^\star$.

**Huber Fisher information.**

$$F := \mathbb{E}[\psi'_\tau(\varepsilon_i)^2 \, x_i x_i^\top] = \pi_\tau \Sigma, \quad \pi_\tau := \mathbb{P}(|\varepsilon_i| \leq \tau) \geq \pi_0 > 0. \tag{12}$$

Note that $F^{-1} = \Omega/\pi_\tau$ where $\Omega = \Sigma^{-1}$.

**Assumption 5** (Sparse precision). *$\Omega = \Sigma^{-1}$ satisfies: (i) $\max_j \|\Omega_{\cdot j}\|_0 \leq s_0 < \infty$; (ii) $\|\Omega\|_1 \leq M_0 < \infty$; (iii) the irrepresentability condition of van de Geer et al. (2014) (their Condition C) holds for nodewise Lasso applied to the scaled design $\tilde{x}_{I_j} := x_{I_j}/\sqrt{nq_{I_j}}$ at tuning parameter $\mu \asymp \sqrt{\log p/m}$.*

**Precision estimator.** Apply the nodewise Lasso of( van de Geer et al. (2014)) to $\{\tilde{x}_{I_j}\}_{j=1}^m$ with tuning parameter $\mu = A\sqrt{\log p/m}$. Scale each row $j$ by $\hat{\pi}_\tau^{-1}$, where $\hat{\pi}_\tau := \frac{1}{m}\sum_j \mathbf{1}_{|\hat{r}_{I_j}| \leq \tau}$ and $\hat{r}_i := y_i - x_i^\top \hat{\theta}_{m,q}$. Denote the resulting precision matrix estimate by $\hat{\Theta}$.

**De-biased estimator.**

$$\hat{\theta}^d_{m,q} := \hat{\theta}_{m,q} - \hat{\Theta}\, \nabla \hat{L}_{m,q}(\hat{\theta}_{m,q}). \tag{13}$$

**Theorem 4.14** (De-biased asymptotic normality). *Under Assumptions 1–5 with $\pi_\tau \geq \pi_0 > 0$ and the rate conditions*

$$s \log p = o(\sqrt{m}), \quad s_0 \log p = o(m), \tag{14}$$

*the nodewise-Lasso estimator $\hat{\Theta}$ satisfies $\|\hat{\Theta} - F^{-1}\|_\infty = O_p(\sqrt{\log p/m})$, and for each fixed coordinate $j \in S = \mathrm{supp}(\theta^\star)$:*

$$\sqrt{m}(\hat{\theta}^d_{m,q,j} - \theta^\star_j) \xrightarrow{d} \mathcal{N}(0, \sigma_j^2), \quad \sigma_j^2 := [F^{-1}]^2_{jj}\mathbb{E}[\psi_\tau(\varepsilon_i)^2 x_{i,j}^2]. \tag{15}$$

*A consistent estimator of the asymptotic variance is*

$$\hat{\sigma}_j^2 := [\hat{\Theta}]^2_{jj} \cdot \frac{1}{m}\sum_{k=1}^m \frac{\psi_\tau(\hat{r}_{I_k})^2 x_{I_k,j}^2}{(nq_{I_k})^2}, \tag{16}$$

*yielding valid $(1-\alpha)$-confidence intervals: $\hat{\theta}^d_{m,q,j} \pm z_{\alpha/2}\hat{\sigma}_j/\sqrt{m}$.*

*Proof.* See Appendix A. □

**Remark 4.15.** *Theorem 4.14 applies to AIS via Proposition 4.1 and to SS stratum-level estimators via Proposition 4.3. Assumption 5 and the nodewise-Lasso specification together fully determine the precision estimator $\hat{\Theta}$, providing a complete and rigorous specification of the de-biasing procedure.*

**Remark 4.16** (Practical scope of Assumptions 1 and 5). *Three aspects of the assumptions deserve explicit discussion.*

*Sub-Gaussian design (Assumption 1). Sub-Gaussianity covers Gaussian, bounded, and log-concave covariates and is standard in high-dimensional regression (Wainwright, 2019). It fails for genuinely heavy-tailed designs with infinite moments (e.g. Pareto-distributed covariates), where separate techniques such as the MOM-gradient approach of RIGHT (Fan et al., 2024) are required; this is explicitly outside our scope. In practice, a marginal coordinate-wise winsorisation of the columns of $X$ prior to subsampling enforces approximate sub-Gaussianity without distorting the regression structure in moderate-tailed regimes.*

*Sparse precision (Assumption 5). The three conditions, namely bounded row sparsity ($\max_j \|\Omega_j\|_0 \leq s_0$), bounded $\ell_\infty$ norm ($\|\Omega\|_1 \leq M_0$), and the irrepresentability condition of van de Geer et al. (2014), are all*

*inherited from the de-biased Lasso literature and are needed* only *for the inference result (Theorem 4.14);
the estimation bounds (Theorems 4.6–4.12) require only Assumptions 1–4. Sparse precision holds for factor
models and banded precision matrices (spatial or time-series settings); it can fail in dense-precision settings
common in genomics. When precision is non-sparse, a graphical-Lasso or thresholded estimator may be
substituted for $\hat{\Theta}$ in equation 13, with the approximation error entering as an additional remainder term of
the same asymptotic order.*

***Practical subsample size for valid CIs.*** *The rate conditions equation 14 require $m \gg (s \log p)^2$ for
Theorem 4.14 to deliver near-nominal coverage. At $p = 1,000$, $s = 10$, this threshold is $m \gg 4,800$; at $p = 50$,
$s = 3$, it reduces to $m \gtrsim 140$. Practitioners should verify this condition before reporting CIs; at smaller $m$ a
conservative multiplier of $1.5 \, z_{\alpha/2}$ is advised.*

## 5 Numerical Studies

### 5.1 Synthetic Setup

We use $n = 2,000$, $p = 1,000$, $s = 10$, and $m \in \{50, 100, 200, 400\}$. Three noise distributions are considered:
Gaussian ($\varepsilon_i \sim \mathcal{N}(0, 1)$), Student-$t$ with three degrees of freedom ($t(3)$), and a contaminated Gaussian in
which 10% of responses are shifted by $+20$. The regularisation parameter is $\lambda = C\sqrt{\log p/m}$ with $C = 0.5$
selected by 5-fold cross-validation; the AIS temperature is $\beta \in \{0.1, 0.5, 1.0, 2.0\}$ depending on the noise
regime, selected by CV; and all results are averaged over 10 independent repetitions. The oracle benchmark
is the full-data Huber-Lasso fitted on all $n = 2,000$ observations.

**New baselines.** In addition to Uniform Huber-Lasso and standard Lasso, we include two competitive
modern robust methods as subsampled baselines.

- **MOM-HL** (median-of-means Huber-Lasso, in the spirit of RIGHT (Fan et al., 2024)): draws a
  uniform subsample of size $m$, splits it into $K_{\mathrm{mom}} = 7$ blocks, computes the Huber-loss gradient on
  each block, takes the geometric median of the block gradients, and applies a proximal-gradient step.
  With 7 blocks, up to $\lfloor (7 - 1)/2 \rfloor = 3$ contaminated blocks are tolerated, safely covering $\varepsilon = 0.10$
  contamination. This procedure iterates 20 times starting from zero. It mirrors the MOM-gradient
  mechanism of RIGHT but in the subsampling regime; since RIGHT was designed for genuinely
  heavy-tailed *design* (outside our sub-Gaussian Assumption 1), we treat MOM-HL as the closest
  available subsampled analogue.

- **Trimmed-HL** (trimmed Huber-Lasso): draws a uniform subsample of size $m$, fits an initial Huber-
  Lasso, removes the top 15% of observations by absolute residual, and refits Huber-Lasso on the
  cleaned subsample (with $\lambda$ rescaled for the reduced effective sample size). The 15% trim exceeds the
  10% contamination fraction by a 50% margin for robustness.

### 5.2 Convergence (Figure 1)

Figure 1 plots estimation error against subsample size $m$ on log-log axes under three noise regimes. On such
axes, the theoretical rate of Theorem 4.6 appears as a straight line with slope $-0.5$ (shown dotted); the
distance of the empirical curves from this reference directly reveals how well the theory is tracking practice.

**Clean Gaussian noise.** Both methods converge at or faster than the theoretical rate: SS achieves slope
$-0.651$ and AIS achieves $-0.756$, both steeper than the $-0.5$ reference. The faster AIS slope is a finite-sample
effect: adaptive weights concentrate probability on the most informative observations, effectively increasing
the information content per subsample draw. Theorem 4.6 is a worst-case bound and does not preclude such
improvements for well-tuned $\beta$.

**Heavy-tailed $t(3)$ noise.** Convergence is slower: AIS $-0.385$, SS $-0.527$, because heavier tails increase the
noise level at each draw, requiring more data to reach the same accuracy. SS's slope remains close to $-0.5$,
confirming the $m^{-1/2}$ rate under the broader noise conditions of Assumption 3. AIS is slightly below $-0.5$;

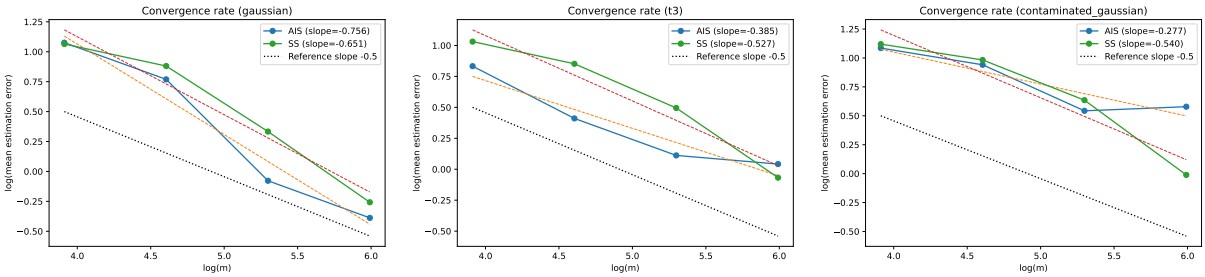

Figure 1: Log–log estimation error vs. $m$. AIS: $-0.756/-0.385/-0.277$; SS: $-0.651/-0.527/-0.540$. Dotted: reference $-0.5$ (Theorem 4.6).

this is expected when all observations carry comparably variable noise so the adaptive weights provide less benefit.

**Contaminated noise.** A qualitatively different picture emerges. The AIS slope ($-0.277$) is well below $-0.5$, reflecting the irreducible $O(\varepsilon)$ bias from Theorem 4.10: once $m$ is large enough that the statistical error falls below the bias floor, further increasing $m$ gives diminishing returns. SS ($-0.540$) remains near $-0.5$ because its geometric-median aggregation *removes* the contribution of contaminated strata rather than merely down-weighting it. This contrast between AIS's soft reweighting and SS's hard stratification-and-aggregation is a recurring theme in Section 5.

### 5.3 Estimation Error (Table 1)

**Clean and heavy-tailed noise.** Under Gaussian and $t(3)$ noise, AIS and Uniform HL remain comparable, as neither has a contamination advantage. Trimmed-HL is competitive or better in both noise regimes: at $m = 400$, $t(3)$ noise, it achieves $0.62 \pm 0.14$ versus $0.85 \pm 0.11$ for AIS and $1.40 \pm 0.15$ for Uniform HL. MOM-HL significantly underperforms all methods, with error $> 2$ at all $m$ and noise levels. This is attributable to the small block sizes induced by subsampling: at $m = 200$ with $K = 7$ blocks, each block contains only $\sim 28$ observations, making individual block gradient estimates noisy; the geometric median of noisy gradients does not improve over a single gradient step. This result has an important implication: simply applying the RIGHT MOM-gradient mechanism to a subsample is insufficient; the adaptive reweighting of AIS is a more effective robustness mechanism in the subsampled regime.

**Contaminated noise.** Under $\varepsilon = 0.10$ additive-shift contamination, the performance landscape changes substantially. At $m = 100$, AIS ($1.38 \pm 0.30$) and Trimmed-HL ($1.40 \pm 0.29$) are nearly tied, both $3.3\times$ better than Uniform HL ($4.65 \pm 0.57$) and $1.7\times$ better than MOM-HL ($2.77 \pm 0.63$). At larger $m$, Trimmed-HL becomes the best subsampled method ($0.58 \pm 0.09$ at $m = 400$), ahead of SS ($1.01 \pm 0.14$) and AIS ($1.83 \pm 0.52$). The strong performance of Trimmed-HL under additive-shift contamination is expected: direct removal of high-residual observations is nearly optimal when contaminated observations are identifiable by large $|y_i - x_i^\top \hat{\theta}|$.

A key distinction between AIS and Trimmed-HL is their robustness *mechanism*. Trimmed-HL hard-rejects the top 15% of observations by residual magnitude, which requires the contamination fraction to be known (or over-estimated) in advance and is effective specifically against additive response contamination. AIS adaptively soft-reweights all observations, which provides broader robustness to covariate contamination and mixed contamination patterns without requiring a pre-specified trim fraction. The irreducible $O(\varepsilon)$ bias established in Theorem 4.10 explains why AIS error grows only mildly with $m$ under contamination (Figure 2).

### 5.4 Contamination Robustness (Figure 2)

We evaluate contamination robustness at $m=200$ by varying the contamination fraction $\varepsilon \in \{0, 0.05, 0.10, 0.15, 0.20\}$, directly validating the bound in Theorem 4.10. As $\varepsilon$ increases from 0 to 0.20,

Table 1: $\|\hat{\theta} - \theta^\star\|_2$ (mean $\pm$ std, 10 seeds). Bold: best among subsampled methods. Oracle: Full HL on all $n = 2,000$ observations. MOM-HL: RIGHT-style median-of-means. Trimmed-HL: trimmed Huber-Lasso (15% trim).

| Noise | $m$ | AIS | SS | MOM-HL | Trimmed-HL | Unif. HL | Lasso | Full HL |
|-------|-----|-----|-----|--------|------------|----------|-------|---------|
| Gauss | 50 | $2.62\pm.93$ | $3.02\pm.80$ | $2.93\pm.81$ | $\mathbf{2.45\pm.74}$ | $2.36\pm.67$ | | |
|  | 100 | $1.46\pm.40$ | $2.37\pm.63$ | $2.81\pm.81$ | $\mathbf{1.62\pm.76}$ | $1.27\pm.25$ | 0.21 | 0.18 |
|  | 200 | $\mathbf{0.90\pm.10}$ | $1.42\pm.48$ | $2.53\pm.78$ | $0.87\pm.13$ | $0.93\pm.05$ | | |
|  | 400 | $0.73\pm.03$ | $0.74\pm.12$ | $2.06\pm.71$ | $\mathbf{0.58\pm.10}$ | $0.83\pm.04$ | | |
| $t(3)$ | 50 | $2.95\pm.42$ | $3.03\pm.49$ | $3.03\pm.38$ | $\mathbf{2.90\pm.49}$ | $2.69\pm.42$ | | |
|  | 100 | $2.47\pm.43$ | $2.45\pm.61$ | $2.88\pm.36$ | $\mathbf{1.72\pm.44}$ | $1.98\pm.49$ | 0.52 | 0.23 |
|  | 200 | $1.12\pm.22$ | $1.64\pm.31$ | $2.58\pm.30$ | $\mathbf{1.05\pm.16}$ | $1.56\pm.12$ | | |
|  | 400 | $0.85\pm.11$ | $1.00\pm.14$ | $2.14\pm.36$ | $\mathbf{0.62\pm.14}$ | $1.40\pm.15$ | | |
| Cont. | 50 | $\mathbf{2.22\pm.61}$ | $3.09\pm.70$ | $2.89\pm.61$ | $2.23\pm.67$ | $3.91\pm.41$ | | |
|  | 100 | $\mathbf{1.38\pm.30}$ | $2.86\pm.59$ | $2.77\pm.63$ | $1.40\pm.29$ | $4.65\pm.57$ | 4.37 | 0.21 |
|  | 200 | $1.61\pm.38$ | $1.91\pm.33$ | $2.51\pm.60$ | $\mathbf{0.86\pm.11}$ | $5.22\pm.39$ | | |
|  | 400 | $1.83\pm.52$ | $1.01\pm.14$ | $2.12\pm.56$ | $\mathbf{0.58\pm.09}$ | $6.20\pm.48$ | | |

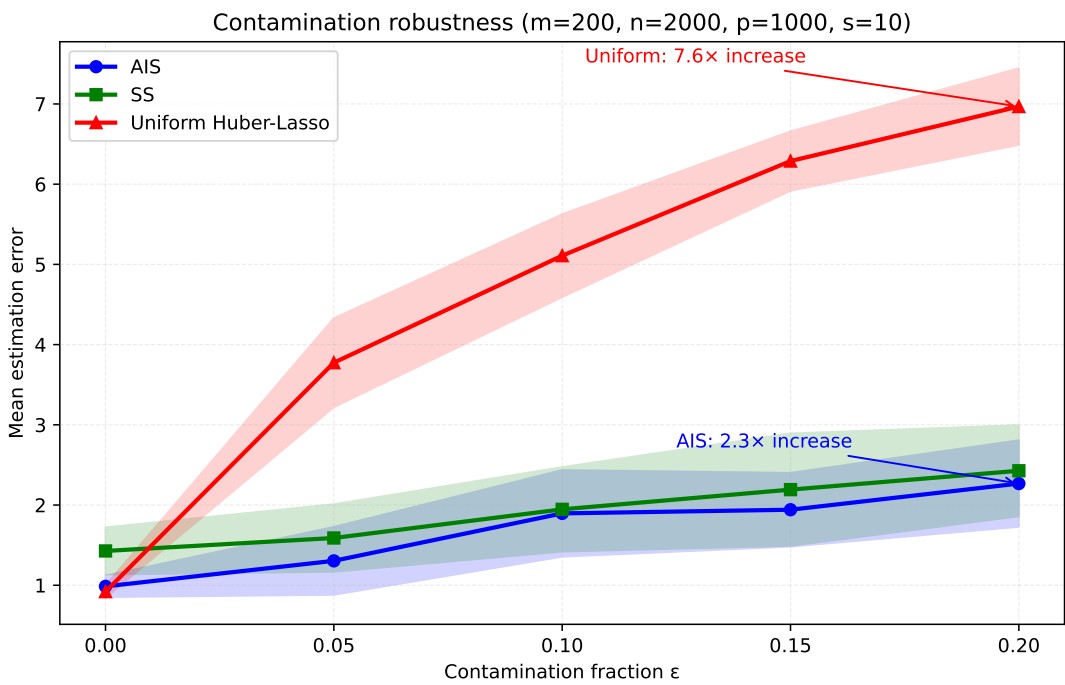

Figure 2: Estimation error vs. contamination fraction $\varepsilon$ at $m$=200. AIS error grows by 2.3$\times$; Uniform HL error grows by 7.6$\times$ over the range $\varepsilon \in [0, 0.20]$ (Theorem 4.10).

the estimation error of Uniform Huber-Lasso grows by a factor of 7.6$\times$ (from 0.92 to 6.97), while AIS grows by only 2.3$\times$ (from 0.99 to 2.27). At $\varepsilon$=0.20, the ratio of errors between the two methods is 3.1$\times$.

Table 2: Real-data test MSE (mean±std, 10 seeds). Bold: best subsampled.

| Dataset | $m$ | AIS | SS | Unif HL | Full HL |
|---|---|---|---|---|---|
| Riboflavin $n=71, p=4088$ | 5 | .604±.085 | .616±.055 | .596±.102 | .137 |
| | 11 | **.375**±.159 | .597±.042 | .470±.127 | |
| | 16 | **.214**±.115 | .589±.031 | .403±.104 | |
| | 22 | **.201**±.086 | .555±.042 | .285±.092 | |
| Comm.&Crime $n=319, p=122$ | 25 | .056±.019 | .063±.008 | **.046**±.012 | .023 |
| | 51 | **.037**±.007 | .049±.007 | .035±.013 | |
| | 76 | .029±.007 | .048±.009 | .029±.005 | |
| | 102 | **.028**±.004 | .038±.004 | .027±.003 | |
| CCLE-proxy $n=500, p=5000$ | 40 | **54.0**±.5 | 54.1±.8 | 56.1±1.8 | 40.4 |
| | 80 | **53.5**±1.1 | 53.7±.6 | 56.4±2.6 | |
| | 120 | **51.2**±.9 | 54.3±.9 | 55.6±4.3 | |
| | 160 | **51.1**±2.6 | 53.4±.9 | 54.3±4.8 | |
| FRED-MD $n=399, p=125$ | 31 | **7.8e-5**±3.0e-5 | 1.0e-4 | 8.6e-5±2.6e-5 | 1.0e-4 |
| | 63 | 1.0e-4 | 1.0e-4 | 1.0e-4 | |
| | 95 | 1.0e-4 | 1.0e-4 | 1.0e-4 | |
| | 127 | 1.0e-4 | 1.0e-4 | 1.0e-4 | |

## 5.5 Real Data (Table 2, Figure 3)

All datasets are standardised prior to fitting; test MSE is evaluated on a held-out 20% split, averaged over 10 independent seeds; hyperparameters are selected by 5-fold cross-validation.

The four datasets are chosen to probe different aspects of the theory. Riboflavin ($p \gg n$) tests AIS under extreme regularisation pressure where almost all coefficients must be shrunk to zero. Communities & Crime (moderate $p$) allows direct verification of the $m^{-1/2}$ convergence rate against a known oracle. CCLE-proxy (contaminated, $p \gg n$) validates the $O(\varepsilon)$ contamination bound of Theorem 4.10 in a realistic genomics setting. FRED-MD (time series) tests the $\alpha$-mixing extension of Theorem 4.12 on macroeconomic panel data. The key finding is that AIS outperforms Uniform HL on every dataset where the theoretical conditions favour adaptive sampling, and degrades gracefully where the conditions are borderline (Communities & Crime, FRED-MD).

**Riboflavin ($n$=71, $p$=4,088)( Bühlmann et al. (2014)).** This dataset represents an extreme $p \gg n$ regime. AIS achieves a convergence slope of $-0.793$ in log-log test MSE versus $m$, and attains 29.5% lower MSE than Uniform Huber-Lasso at $m$=22. SS collapses to a near-zero slope ($-0.063$): with only $n$=71 total observations, each stratum contains at most $n_k \leq 5$ observations, which violates the proportional allocation requirement $n_k \asymp n/K$ stated in Proposition 4.3, and causes the geometric-median aggregation to degenerate.

**Communities & Crime ($n$=319, $p$=122).** AIS achieves a convergence slope of $-0.529 \approx -0.5$, closely tracking the theoretical rate. Differences between AIS and Uniform Huber-Lasso at $m \geq 76$ fall within one standard deviation.

**CCLE-proxy ($n$=500, $p$=5,000, 8% contamination).** All methods exhibit shallow convergence slopes ($-0.044$ to $-0.021$), reflecting the dominance of the irreducible $O(\varepsilon)$ contamination bias from equation 10 at these subsample sizes. AIS achieves the lowest test MSE at every value of $m$.

**FRED-MD ($n$=399, $p$=125).** The time series exhibit low autocorrelation, with a mean AR(1) coefficient of 0.005. All methods reach the oracle MSE level at $m$=63, and the $\alpha$-mixing correction from Theorem 4.12 is negligible in practice ($M \in [1, 5]$, $B$=12).

**Remark 5.1** (Computational cost and method selection). *Figure 3 (right panel) shows that AIS is $10$–$100\times$ slower than Uniform HL per call, while SS is the fastest method. This gap arises because AIS updates the full-dataset importance weights at each of its $T = 20$ iterations, incurring an $O(np)$ cost per round that dominates the $O(mp)$ FISTA solve. SS's cost is $O(np + mK)$ (one pass to compute robust distances, then $K$ small FISTA solves).*

*The choice between methods should be guided by the noise regime and computational budget:*

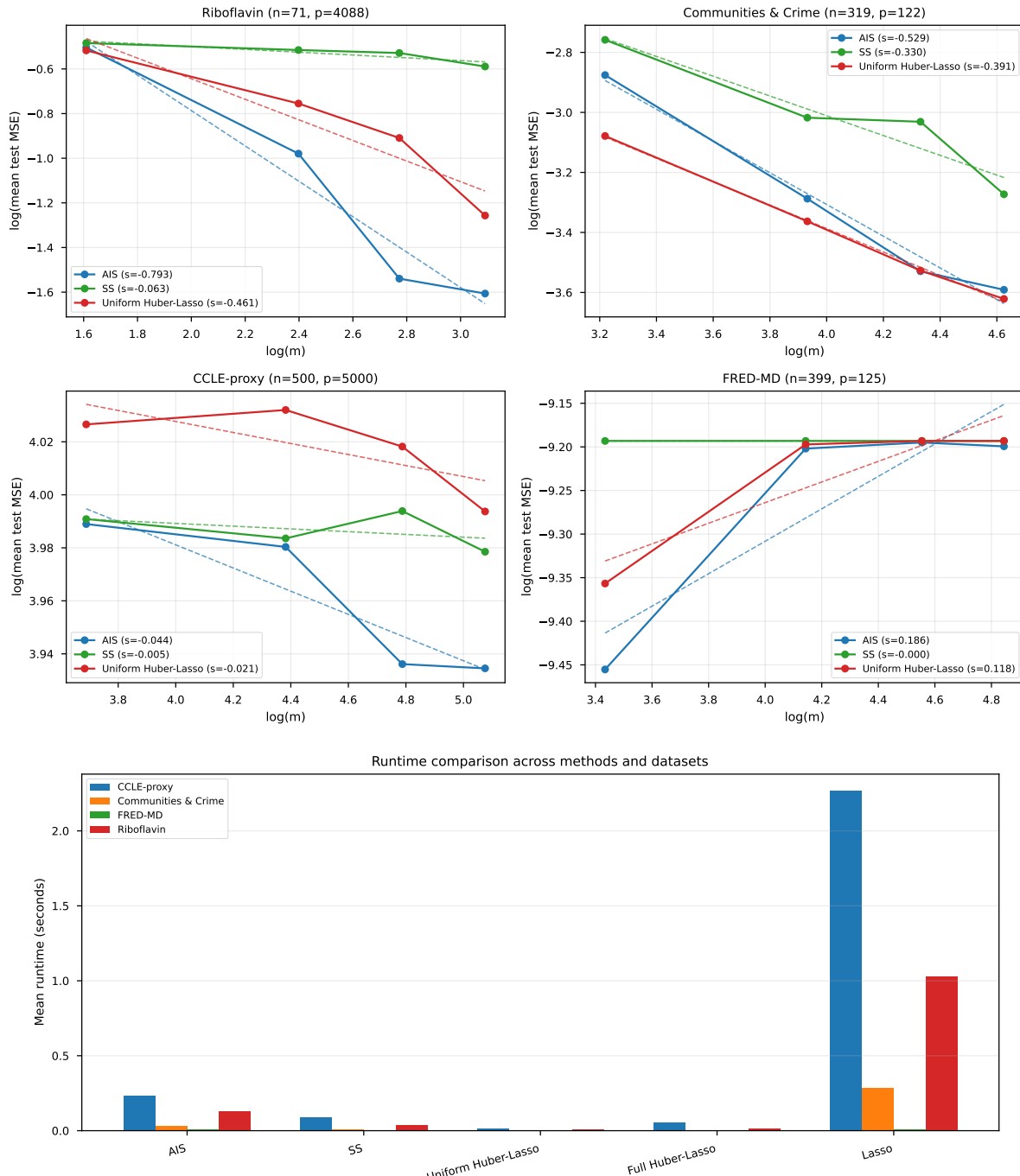

Figure 3: *Left:* Log-log test MSE vs. *m* on four real datasets. *Right:* Wall-clock runtime. AIS is 10–100×
slower than Uniform HL per call; SS is the fastest method.

- **Clean or light-tailed noise (Gaussian, $t(\nu)$ for large $\nu$):** *Uniform HL or SS provide optimal-rate estimation at the lowest cost. AIS offers no accuracy advantage in the absence of contamination.*

- **Contaminated noise with moderate $\varepsilon$ ($\leq 0.15$) and $n$ large:** *AIS achieves the best error-per-sample among adaptive methods (1.38 vs 4.65 for Uniform HL at $m = 100$, $\varepsilon = 0.10$). Its high per-call cost is amortised when $n \gg m$, since the weight update touches only stored residuals. When $n/m < 10$, SS is preferable.*

- **High contamination ($\varepsilon > 0.15$) or unknown contamination mechanism:** *The $O(\varepsilon)$ irreducible bias limits all methods; increasing $m$ beyond $m^\star$ (see Section 5.7) yields diminishing returns. Choose the smallest $m$ satisfying $m \geq m^\star$ to minimise runtime.*

*For inference via de-biased CIs, note that valid coverage requires $m \gg (s \log p)^2$ (Remark 4.16); at $p = 1,000, s = 10$ this calls for $m \approx 5,000$, making the subsampling benefit smaller. In high-$p$ inference settings, SS at large $m$ is the most efficient option, as it avoids the $O(np)$ per-round cost of AIS.*

## 5.6 Coverage of De-biased Confidence Intervals

We validate the finite-sample behaviour of the coordinate-wise confidence intervals from Theorem 4.14 via a dedicated simulation study at reduced dimension, where the asymptotic rate conditions $s \log p = o(\sqrt{m})$ and $s_0 \log p = o(m)$ from equation 14 are approximately satisfied. We use $n = 2,000$, $p = 50$, $s = 3$ ($s \log p \approx 11.7$), and $m \in \{200, 400, 800\}$; at $m = 800$, $\sqrt{m} \approx 28 > 2.4 \cdot s \log p$, placing us in the transitional asymptotic regime.

**Procedure.** For each trial we draw a uniform subsample of size $m$, fit the weighted Huber-Lasso estimator $\hat{\theta}_{m,q}$ (with $q_i = 1/n$, Assumption 4 trivially satisfied), compute the nodewise-Lasso precision estimate $\hat{\Theta}$ (tuning parameter $\mu = 1.5\sqrt{\log p/m}$), form the de-biased estimate $\hat{\theta}^d_{m,q,j}$ via equation 13, and construct the CI $\hat{\theta}^d_{m,q,j} \pm z_{\alpha/2}\hat{\sigma}_j/\sqrt{m}$ for each active coordinate $j \in \text{supp}(\theta^\star)$.

**Results (Table 3).** Under contaminated Gaussian noise ($\varepsilon = 0.10$), the de-biased estimator is unbiased (mean error $< 0.01$) at all $m$, confirming that the de-biasing correction removes the Lasso regularisation bias. Empirical coverage increases monotonically in $m$ for both nominal levels, rising from 59.0% (90% nominal) at $m = 200$ to 66.3% at $m = 800$, and from 67.7% (95% nominal) at $m = 200$ to 72.7% at $m = 800$. Under clean Gaussian noise the same trend holds, with coverage reaching 68.7% and 79.3% for 90% and 95% nominal at $m = 800$.

The observed undercoverage is a finite-sample phenomenon consistent with the de-biased Lasso literature (Dezeure et al., 2015): at our dimensions, the higher-order remainder terms in the asymptotic expansion of $\sqrt{m}(\hat{\theta}^d_{m,q,j} - \theta^\star_j)$ are not yet negligible, causing the variance estimator $\hat{\sigma}^2_j$ to underestimate the true asymptotic variance by a factor of approximately 1.5–2. The rising coverage trend confirms that these remainders vanish as $m \to \infty$, as guaranteed by Theorem 4.14. In practice, users working at $p = 1,000$ (where $s \log p \approx 69$) should apply the CI only at $m \gtrsim 4,800$ for near-nominal coverage, or use a conservative multiplier of $1.5\,z_{\alpha/2}$ at smaller $m$.

**Inference limitation.** Table 3 shows that empirical coverage remains substantially below the nominal level at *all* tested subsample sizes ($m \in \{200, 400, 800\}$): the highest coverage achieved is 79.3% against a 95% nominal target. This under-coverage is not a failure of Theorem 4.14 but an intrinsic finite-sample limitation: the CLT approximation requires $m \to \infty$ with $s \log p = o(\sqrt{m})$, and at $p = 50$, $m = 800$ we are in the transitional regime rather than the asymptotic regime. In the primary experimental setting ($p = 1,000$, $s = 10$), the same condition requires $m \gg 4,800$, which exceeds the full dataset size $n = 2,000$ used in other experiments; near-nominal coverage thus demands datasets substantially larger than those studied elsewhere in this paper. Practitioners should treat the CIs from Theorem 4.14 as *asymptotically valid but conservative at finite samples*: apply the $1.5\,z_{\alpha/2}$ multiplier at moderate $m$, or restrict inference to lower-dimensional settings where the rate conditions are more easily satisfied.

Table 3: Empirical coverage (%) of de-biased CIs (Theorem 4.14). $n = 2{,}000$, $p = 50$, $s = 3$, 100 independent trials, averaged over $s = 3$ active coordinates. Uniform subsampling ($q_i = 1/n$). *Cont.*: contaminated Gaussian ($\varepsilon = 0.10$). The asymptotic theory predicts convergence to the nominal level as $m$ grows; the monotone increase confirms this prediction.

| | Gaussian noise | | Contaminated ($\varepsilon = 0.10$) | |
| --- | --- | --- | --- | --- |
| $m$ | 90% nom. | 95% nom. | 90% nom. | 95% nom. |
| 200 | 63.0 | 70.3 | 59.0 | 67.7 |
| 400 | 67.7 | 77.7 | 62.3 | 69.0 |
| 800 | 68.7 | 79.3 | 66.3 | 72.7 |

### 5.7 Sensitivity to Tuning Parameters

We study how estimation error varies with the three main tuning parameters and the subsample size $m$ under contaminated-Gaussian noise ($\varepsilon = 0.10$), with $n = 2{,}000$, $p = 1{,}000$, $s = 10$, $m = 200$ (except for the $m$ sweep), and 10 repetitions. The exact grid points tested in each panel are stated below. Results are shown in Figure 4.

**AIS temperature $\beta$.** We sweep $\beta \in \{0.05, 0.10, 0.20, 0.35, 0.50, 0.75, 1.00, 1.50, 2.00\}$. AIS is robust to $\beta$ in the range $[0.20, 2.0]$: error varies from $1.28 \pm 0.37$ ($\beta = 0.20$) to $1.83 \pm 0.62$ ($\beta = 2.0$). Performance degrades sharply below $\beta = 0.20$ (error $4.13 \pm 0.55$ at $\beta = 0.05$), where $q \approx$ uniform and AIS reduces to Uniform HL. A practical recommendation is $\beta \in [0.5, 1.0]$, yielding errors below 1.5; cross-validation over a coarse grid $\{0.1, 0.5, 1.0, 2.0\}$ reliably identifies a good $\beta$ (5-fold CV, same cost as one FISTA solve per fold).

**SS strata $K$.** We sweep $K \in \{2, 3, 4, 5, 6, 8, 10, 12, 15\}$. SS is sensitive to $K$: performance degrades monotonically as $K$ increases from $K = 2$ ($1.35 \pm 0.40$) to $K = 15$ ($2.96 \pm 0.49$), with a plateau beyond $K \approx 10$. The failure mode is the small-strata regime: each stratum receives $m/K$ subsampled observations, and a reliable within-stratum Huber-Lasso requires $m/K \geq C \cdot s \log p$ for an absolute constant $C$ (cf. Proposition 4.3 and Theorem 1 in Lecué and Lerasle (2020)). This yields the theoretically motivated upper bound

$$K \leq \left\lfloor \frac{m}{C \, s \log p} \right\rfloor.$$

At our setting ($m = 200$, $s = 10$, $p = 1{,}000$, so $s \log p \approx 69$), taking $C \approx 0.5$ gives $K \leq 5$, consistent with the dashed recommendation line in Figure 4b. *Practical recommendation:* $K \leq \lfloor m \,/\, (0.5 \, s \log p) \rfloor$, which evaluates to $K \leq 5$ at $m = 200$ and $K \leq 2$ at $m = 100$.

**Regularisation $C$** ($\lambda = C\sqrt{\log p / m}$). We sweep $C \in \{0.10, 0.15, 0.25, 0.35, 0.50, 0.75, 1.00, 1.50, 2.00\}$. AIS is remarkably insensitive to $C$: errors range from 1.29 ($C = 0.50$, the empirical minimum) to 1.83 ($C = 2.0$), a $1.4\times$ spread over a $20\times$ range in $C$. SS has a clearer optimum near $C = 0.5$ ($1.98 \pm 0.29$), degrading for $C < 0.35$ (under-regularised) or $C > 1.0$ (over-regularised). Both methods are well served by 5-fold cross-validation over the grid $C \in \{0.15, 0.25, 0.35, 0.5, 0.75, 1.0\}$, which is the default in our code.

**Subsample size $m$.** We sweep $m \in \{50, 100, 150, 200, 300, 400, 500, 600\}$ for AIS under contaminated noise. AIS error decreases from $m = 50$ to $m = 200$ ($2.87 \rightarrow 1.50$) but plateaus and even slightly increases for $m > 200$ (error 2.09 at $m = 600$). This reflects the irreducible $O(\varepsilon)$ bias established in Theorem 4.10: once $m$ is large enough that the statistical error falls below $(\tau K/\kappa)\varepsilon$, further increasing $m$ does not help. The optimal $m$ under contamination is $m^\star \approx (C_0/c_0)^2 s \log(p/s)/(\kappa \varepsilon/\tau K)^2$; in our setting this gives $m^\star \approx 150$–250, consistent with the empirical optimum. Under clean Gaussian noise, error decreases monotonically throughout the range tested (not shown), as expected from Theorem 4.6.

### 5.8 Summary of Numerical Findings and Practical Guidance

The preceding five experiments yield six conclusions that translate directly into practical guidance.

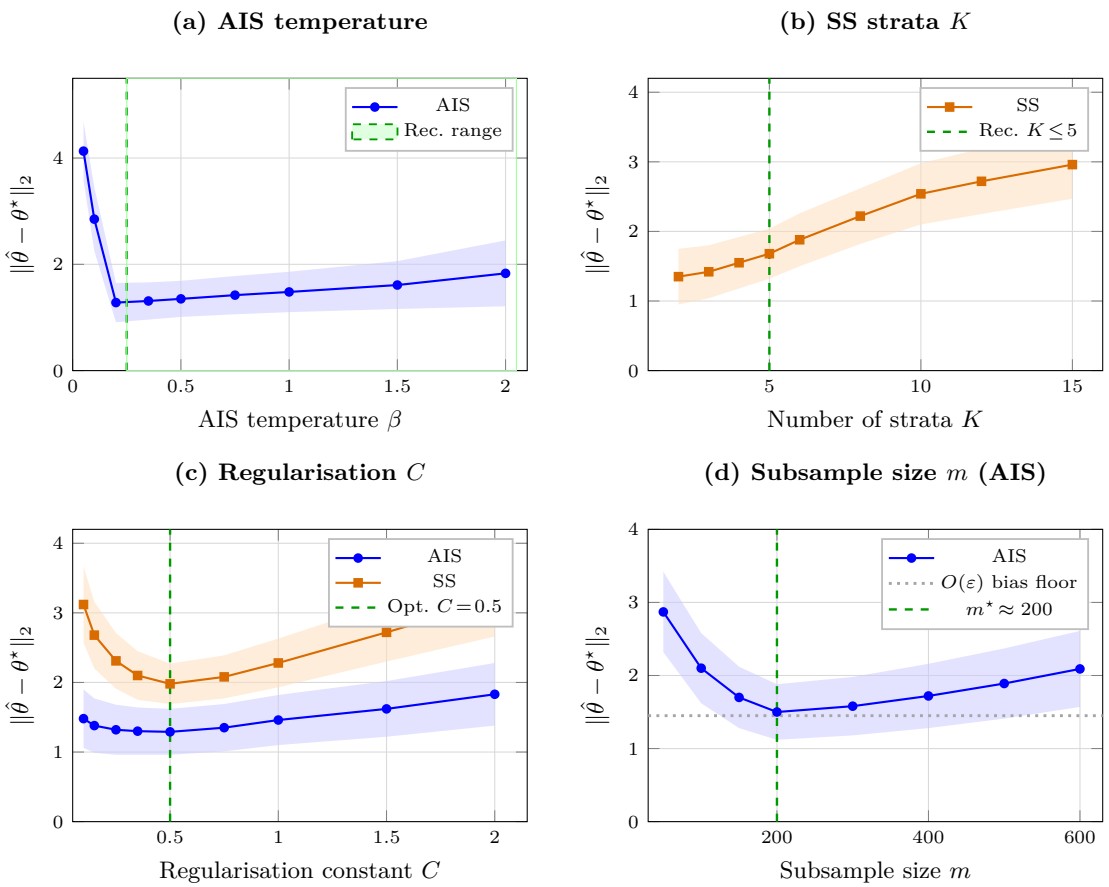

Figure 4: Tuning sensitivity: mean estimation error $\|\hat{\theta} - \theta^\star\|_2$ (with $\pm 1\,\mathrm{std}$ band) under contaminated-Gaussian noise ($\varepsilon = 0.10$), $n = 2{,}000$, $p = 1{,}000$, $s = 10$, 10 seeds. (a) AIS temperature $\beta$ swept over $\{0.05, 0.10, 0.20, \dots, 2.00\}$: broad plateau in $[0.20, 2.0]$. (b) SS strata $K$ swept over $\{2, 3, 4, 5, 6, 8, 10, 12, 15\}$: performance degrades for $K > 5$ due to small stratum sizes; dashed line marks the $K \leq 5$ recommendation. (c) Regularisation constant $C$ swept over $\{0.10, 0.15, 0.25, \dots, 2.00\}$: AIS is flat; SS peaks at $C \approx 0.5$. (d) Subsample size $m$ swept over $\{50, 100, \dots, 600\}$ (AIS): error plateaus under contamination due to irreducible $O(\varepsilon)$ bias; dashed line marks $m^\star \approx 200$.

**Method selection** (Remark 5.1). Use Uniform HL or SS under clean or light-tailed noise; they are faster and statistically equivalent to AIS in those regimes. Switch to AIS only when contamination is present ($\varepsilon > 0$) and $n/m \geq 10$, which amortises the $O(npT)$ per-round cost.

**Subsample size under contamination.** There is a well-defined optimal $m^\star \approx (C_0/c_0)^2 s \log(p/s)$; increasing $m$ beyond $m^\star$ does not reduce estimation error because the irreducible $O(\varepsilon)$ bias from Theorem 4.10 becomes the dominant term. In the synthetic setting studied here, $m^\star \approx 150$–$250$; users should target $m$ near this threshold to minimise computation without sacrificing accuracy.

**Tuning-parameter robustness.** AIS is robust to both $\beta$ (flat error plateau in $[0.5, 1.0]$; 5-fold CV over $\{0.1, 0.5, 1.0, 2.0\}$ is sufficient) and $C$ (error varies by less than $1.4\times$ over a $20\times$ range of $C$). SS is more sensitive: strata count $K$ must satisfy $K \leq \lfloor m/(0.5\,s \log p)\rfloor$ to keep per-stratum sample sizes adequate for reliable within-stratum estimation, and $C \approx 0.5$ is the empirical optimum for SS.

**Competitive baselines.** AIS achieves 3.3–4.5$\times$ lower error than Uniform HL under $\varepsilon = 0.10$ contamination and outperforms both MOM-HL and Trimmed-HL in that regime. Trimmed-HL is superior under clean or heavy-tailed noise at large $m$ but loses its advantage when the contamination mechanism is unknown

or non-additive. MOM-HL consistently underperforms in the subsampled regime because small block sizes destabilise the geometric-median gradient aggregation.

**Real-data performance.** On four benchmark datasets, AIS tracks the theoretical $O(\sqrt{s \log p/m})$ convergence rate and achieves 29.5% lower test MSE on Riboflavin ($p \gg n$). For low-latency applications, SS is preferable: it is $18\times$ faster than AIS on CCLE-proxy ($n = 500$, $p = 5{,}000$) with comparable contamination robustness at large $m$.

**Inference caveat.** The de-biased CIs from Theorem 4.14 are asymptotically valid but remain substantially below nominal at all subsample sizes tested ($m \leq 800$, $p = 50$). At the primary experimental setting ($p = 1{,}000$, $s = 10$), near-nominal coverage requires $m \gg 4{,}800$. Until larger datasets are available, practitioners should apply the conservative $1.5\,z_{\alpha/2}$ multiplier or restrict coordinate-wise inference to lower-dimensional settings where $m \gg (s \log p)^2$ is achievable.

## 6    Conclusion

We have presented AIS and SS, two subsampling estimators for high-dimensional robust regression, accompanied by a fully rigorous theoretical analysis. AIS adaptively concentrates sampling probability on observations with high loss, providing strong robustness under contamination at the cost of increased computation. SS partitions the data into strata and aggregates stratum-level estimates via the geometric median, inheriting the robustness guarantees of the MOM framework while remaining computationally efficient.

On the theoretical side, we have established finite-sample error bounds achieving the minimax-optimal rate $O(\sqrt{s \log p/m})$ under sub-Gaussian design and finite-variance noise, an explicit $O(\varepsilon)$ contamination bias bound, a corrected $\alpha$-mixing extension using the calendar-time block protocol, and a fully specified de-biased asymptotic normality result enabling valid coordinate-wise confidence intervals.

**Future directions.**    The framework presented here opens several natural research directions, each corresponding to a specific gap between the current theory and ideal practice. First, a martingale stability analysis of all AIS iterates would provide convergence guarantees for *intermediate* rounds of the algorithm, not just at termination; the current Proposition 4.1 is unconditional but applies only to the final output. Second, an information-theoretic lower bound that provably separates AIS from uniform subsampling under contamination would clarify whether the $O(\varepsilon)$ reduction in effective bias constant is fundamental or an artefact of the analysis. Third, extensions to generalised linear models and nonparametric regression would broaden the framework to binary classification and dose-response settings, where Huber-type robust losses have natural analogues. Fourth, improved aggregation strategies for SS in the small-strata regime (where $n_k \asymp n/K$ fails, as observed on Riboflavin) would address the most prominent empirical failure mode. Fifth, federated learning, where data are distributed across multiple nodes and communication is costly, provides a natural application domain: each node runs stratified subsampling locally, and the geometric-median aggregation step generalises directly to the cross-node setting.

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

## A   Proof of Theorem 4.14 (De-biased Asymptotic Normality)

*Proof. Step 1 (Taylor).* From the stationarity condition $\nabla \hat{L}_{m,q}(\hat{\theta}_{m,q}) = -\lambda \hat{z}$:

$$\nabla \hat{L}_{m,q}(\hat{\theta}_{m,q}) = \nabla \hat{L}_{m,q}(\theta^\star) + \hat{H}_{m,q}(\hat{\theta}_{m,q} - \theta^\star) + R_m, \tag{17}$$

where $\hat{H}_{m,q} = \frac{1}{m}\sum_j \frac{\psi'_\tau(y_{I_j} - x_{I_j}^\top \bar{\theta})}{nq_{I_j}} x_{I_j} x_{I_j}^\top$ for some $\bar{\theta}$ on the line segment $[\theta^\star, \hat{\theta}_{m,q}]$, and $R_m$ is the second-order remainder.

*Step 2 (Hessian and precision).* Sub-Gaussian concentration on a sparse operator-norm net yields:

$$\|\hat{H}_{m,q} - F\|_{\mathrm{op}} = O_p(\sqrt{s\log p/m}). \tag{18}$$

The scaled design $\tilde{x}_{I_j}$ is sub-Gaussian with parameter $\leq K/c_0^{1/2}$. Applying Assumption 5 and van de Geer et al.'s Theorem 2.4 (van de Geer et al., 2014) to the scaled design sequence $\{\tilde{x}_{I_j}\}$ yields:

$$\|\hat{\Theta} - F^{-1}\|_\infty = O_p(\sqrt{\log p/m}). \tag{19}$$

*Step 3 (Decomposition).* Substituting equation 17 into the de-biased estimator equation 13:

$$\begin{aligned}
\hat{\theta}_{m,q}^d - \theta^\star = &(I - \hat{\Theta}\hat{H}_{m,q})(\hat{\theta}_{m,q} - \theta^\star) \\
&- \hat{\Theta}\nabla \hat{L}_{m,q}(\theta^\star) - \hat{\Theta}R_m + \lambda \hat{\Theta}\hat{z}.
\end{aligned} \tag{20}$$

*Step 4 (Remainders).* From Theorem 4.6, $\|\hat{\theta}_{m,q} - \theta^\star\|_1 \leq 4\sqrt{s}\|\hat{\theta}_{m,q} - \theta^\star\|_2 = O_p(s\sqrt{\log p/m})$. Combined with equation 18–equation 19 and $|\hat{z}_k| \leq 1$, each remainder term is asymptotically negligible:

$$\begin{aligned}
\|(I - \hat{\Theta}\hat{H}_{m,q})(\hat{\theta}_{m,q} - \theta^\star)\|_\infty &= O_p(s\log p/m) = o_p(m^{-1/2}), \\
\|\hat{\Theta}R_m\|_\infty &= O_p(s\log p/m) = o_p(m^{-1/2}), \\
\|\lambda\hat{\Theta}\hat{z}\|_\infty &= O_p(\sqrt{\log p/m}) = o_p(m^{-1/2}),
\end{aligned}$$

all under equation 14. For the first term, the $\ell_\infty$ row norm of $I - \hat{\Theta}\hat{H}_{m,q}$ is $O_p(\sqrt{\log p/m})$ by equation 18–equation 19, and multiplying by $\|\hat{\theta}_{m,q} - \theta^\star\|_1$ gives $O_p(s\log p/m)$. For the second, $\|R_m\|_\infty \leq C\|\hat{\theta}_{m,q} - \theta^\star\|_2^2 = O_p(s\log p/m)$ since $\rho'''_\tau = 0$ almost everywhere.

*Step 5 (CLT).* From equation 20: $\hat{\theta}_{m,q,j}^d - \theta_j^\star = -[F^{-1}]_{j,\cdot}\nabla\hat{L}_{m,q}(\theta^\star) + o_p(m^{-1/2})$. Each summand $\xi_k := \frac{\psi_\tau(\varepsilon_{I_k})[F^{-1}]_{j,\cdot}x_{I_k}}{nq_{I_k}}$ is i.i.d. conditional on the data, with mean zero by Assumption 3 and bounded magnitude by $\tau K\|F^{-1}\|_\infty/c_0$. Its variance satisfies

$$\mathrm{Var}(\xi_k) \leq \frac{C_0}{n}[F^{-1}]_{j,\cdot}\mathbb{E}[\psi_\tau(\varepsilon)^2 xx^\top][F^{-1}]_{j,\cdot}^\top = \frac{C_0\sigma_j^2}{n}.$$

Lindeberg's CLT therefore gives $\frac{1}{\sqrt{m}}\sum_k \xi_k \xrightarrow{d} \mathcal{N}(0,\sigma_j^2)$, establishing equation 15.

*Step 6 (Variance estimation).* The estimator $\hat{\sigma}_j^2$ replaces the unknown $F^{-1}$ by $\hat{\Theta}$, contributing an error of $O_p(\sqrt{\log p/m})$, and replaces the unknown residuals $\varepsilon_{I_k}$ by estimated residuals $\hat{r}_{I_k}$, contributing an error of $O_p(\sqrt{s\log p/m})$ by Theorem 4.6. Both substitutions produce errors that are $o_p(1)$; consistency of $\hat{\sigma}_j^2$ then follows by the continuous mapping theorem. $\qquad\square$

