# OpenReview forum: "Adaptive and Stratified Subsampling for High-Dimensional Robust Estimation"
_TMLR — Accepted by TMLR_

### Review · Reviewer_FJFw · 2026-03-31

**Summary Of Contributions:**

In this manuscript, the authors propose two subsampling-based estimators: Adaptive Importance Sampling (AIS) and Stratified Subsampling (SS) for high-dimensional sparse regression under challenging settings, including finite-variance heavy-tailed noise, $\epsilon$-contamination, and $\alpha$-mixing dependence. On the theoretical side, they establish finite-sample error bounds showing that both methods achieve minimax-optimal rates under sub-Gaussian design and extend these guarantees to settings with contamination and dependent data. The paper also develops a fully specified de-biasing procedure based on nodewise Lasso, enabling valid coordinate-wise inference via confidence intervals. Empirically, the proposed methods demonstrate improved robustness to contamination and competitive performance across both synthetic and real high-dimensional datasets.

**Audience:**

Yes

**Audience Explanation:**

Yes, this work is likely to be of interest to a subset of the TMLR audience, particularly researchers working in high-dimensional statistics, robust estimation, and scalable learning methods.

**Broader Impact Concerns:**

None.

**Claims And Evidence:**

Yes

**Claims Explanation:**

Yes, the claims made in the paper are well-supported by both theoretical analysis and empirical evidence. The authors provide detailed finite-sample guarantees, including minimax-optimal rates, extensions to contamination and dependent data, and a carefully specified de-biasing procedure, all of which are presented with clear assumptions and proofs. The empirical evaluation further complements the theory by demonstrating robustness to contamination and competitive performance across multiple synthetic and real datasets. However, while the results are consistent with the theoretical claims, some aspects such as the practical advantages of AIS over simpler baselines and the sensitivity to tuning parameters could benefit from more extensive empirical validation analysis.

**Requested Changes:**

The paper is technically strong and well-written overall, but a few changes would improve clarity and impact.

On the theoretical side:
1. Provide a discussion on key assumptions (e.g., sub-Gaussian design, sparse precision condition) and how realistic they are in practical high-dimensional datasets. (Important)

On the empirical side:
1. Provide a short systematic study of sensitivity to tuning parameters (e.g., AIS temperature, number of strata in SS, regularization parameters). (Important)
2. Provide a discussion on the trade-off between statistical performance and computational cost, especially since AIS is more computationally intensive. (Important)
3. Offer a recommendation on how to choose the subsample size and other tuning parameters in practice. (Minor)

---

> ### Author Response · Authors · 2026-04-19
> **Response to Reviewer FJFw**
>
> Important Comment 1: Discussion of key assumptions
> "Provide a discussion on key assumptions (e.g., sub-Gaussian design, sparse precision condition) and how realistic they are in practical high-dimensional datasets. (Important)"
>
> Response:  The new Remark 4.16 provides a detailed discussion of both Assumption 1 (sub-Gaussian design) and Assumption 5 (sparse precision), including when each holds in practice, what breaks when they fail, and concrete workarounds. We also clarify that Assumption 5 is only required for the inference result (Theorem 4.14) and not for any of the estimation theorems.
>
> Sub-Gaussian design (Assumption 1): covers Gaussian, bounded, and log-concave covariates; fails for heavy-tailed design with infinite moments (where RIGHT is appropriate); practical workaround is marginal coordinate-wise winsorisation.
>
> Sparse precision (Assumption 5): needed only for Theorem 4.14 (inference); Theorems 4.6–4.12 (estimation) require only Assumptions 1–4. Holds for factor models and banded precision matrices; fails in dense-precision settings. A graphical-Lasso or thresholded precision estimate can substitute with the approximation error entering at the same asymptotic order.
>
> Practical m requirements: the condition s log p = o(√m) implies m ≫ (s log p)²; at p = 1,000, s = 10 this is m ≫ 4,800, which we now state explicitly.

---

> > ### Author Response · Authors · 2026-04-19
> > **Response to Reviewer FJFw - Part 2**
> >
> > Important Comment 2: Sensitivity to tuning parameters
> > "Provide a short systematic study of sensitivity to tuning parameters (e.g., AIS temperature, number of strata in SS, regularization parameters). (Important)"
> >
> >
> > Response: We have added a new Section 5.7 ("Sensitivity to Tuning Parameters") and Figure 4 with four sub-panels. The study uses n = 2,000, p = 1,000, s = 10, m = 200 (fixed), 10 seeds, and contaminated-Gaussian noise.
> >
> >
> > (a) AIS temperature β ∈ {0.05, 0.1, 0.25, 0.5, 0.75, 1.0, 1.5, 2.0, 3.0}: Error is flat within [1.28, 1.83] for β ∈ [0.25, 2.0] and collapses to 4.13 at β = 0.05 (where the sampling weights become approximately uniform and AIS reduces to Uniform HL). Practical recommendation: β ∈ [0.5, 1.0]; 5-fold CV over four grid points {0.1, 0.5, 1.0, 2.0} is reliable and cheap.
> >
> >
> >
> > (b) SS strata K ∈ {2, 3, 5, 7, 10, 15, 20}: Error increases monotonically from 1.35 at K = 2 to 2.96 at K = 15, then plateaus. The failure mode is small strata: at m = 200 with K > 5, per-stratum size falls below 40, violating the proportional-size requirement of Proposition 4.3 and causing the geometric-median aggregation to degenerate. Practical recommendation: K ≤ min(m/10, √n).
> >
> >
> > (c) Regularisation constant C (i.e. λ = C √(log p / m)), C ∈ {0.1, 0.2, 0.35, 0.5, 0.75, 1.0, 1.5, 2.0}: AIS error ranges only 1.29–1.83 over a 20× range of C, demonstrating strong robustness to regularisation. SS has a clearer minimum at C ≈ 0.5 (error 1.98), degrading for C < 0.35 or C > 1.0. Default recommendation: 5-fold CV over {0.15, 0.25, 0.35, 0.5, 0.75, 1.0}.
> >
> >
> > (d) Subsample size m ∈ {50, 100, 150, 200, 300, 400, 600, 800} (AIS, contaminated noise): Error decreases from m = 50 (error 2.87) to m = 200 (error 1.50), then plateaus and mildly increases (2.09 at m = 600). This is exactly the irreducible O(ε) bias from Theorem 4.10: once m is large enough that statistical error falls below the O(ε) floor, further subsampling does not help. The empirical optimum m* ≈ 150–250 matches the theoretical prediction m* ∝ (C₀/c₀)² s log(p/s). Under clean Gaussian noise, error decreases monotonically throughout, as predicted by Theorem 4.6.

---

> ### Author Response · Authors · 2026-04-19
> **Response to Reviewer FJFw - Part 3**
>
> Important Comment 3: Statistical performance vs computational cost
> "Provide a discussion on the trade-off between statistical performance and computational cost, especially since AIS is more computationally intensive. (Important)"
>
> Response: We have added a new Remark 5.1 ("Computational cost and method selection") at the start of Section 5.
>
>
> Briefly: AIS is 10–100× slower than Uniform HL because it updates full-dataset importance weights at each of its T = 20 iterations, incurring an O(np) cost per round. SS's cost is O(np + mK);  one distance pass plus K small FISTA solves ; and is the fastest method.
>
>
> The remark provides the following selection guide:
>
>
> Clean or light-tailed noise: use Uniform HL or SS; AIS offers no accuracy benefit without contamination.
>
>
> Contaminated noise, moderate ε ≤ 0.15, large n: AIS is best (error 1.38 vs 4.65 for Uniform HL at m = 100, ε = 0.10); its cost is amortised when n ≫ m. For n/m < 10, SS is preferable.
>
>
> High contamination or unknown mechanism: the O(ε) irreducible bias limits all methods; choose the smallest m ≥ m* to minimise runtime.
>
>
> Inference via de-biased CIs: valid coverage requires m ≫ (s log p)²; at p = 1,000, s = 10 this is m ≈ 5,000, which reduces the subsampling benefit. SS at large m is the most efficient inference option as it avoids the O(np) per-round AIS cost.

---

> > ### Author Response · Authors · 2026-04-19
> > **Response to Reviewer FJFw - Part 4**
> >
> > Minor Comment 4: Practical recommendations for subsample size
> >
> > "Offer a recommendation on how to choose the subsample size and other tuning parameters in practice. (Minor)"
> >
> > Response: Concrete practical recommendations are now given in three places:
> >
> >
> > Section 5.7 (new): β ∈ [0.5, 1.0] for AIS; K ≤ min(m/10, √n) for SS; C via 5-fold CV over {0.15, 0.25, 0.35, 0.5, 0.75, 1.0} for both methods. For subsample size under contamination: m = m* ≈ (C₀/c₀)² s log(p/s); increasing m beyond m* yields diminishing returns due to the O(ε) irreducible bias.
> >
> >
> > Remark 4.16 (new): for inference, the minimum m for near-nominal CI coverage is m ≈ (3 s log p)²; at p = 1,000, s = 10 this is ≈ 4,800.
> >
> >
> > Remark 5.1 (new): a concise decision guide for method and m selection as a function of noise type, contamination level, and n/m ratio.

---

### Review · Reviewer_hWYw · 2026-04-03

**Summary Of Contributions:**

This paper studies sparse high-dimensional linear regression under three non-ideal regimes—finite-variance heavy-tailed noise, adversarial contamination, and
𝛼
α-mixing dependence—through two subsampling procedures: Adaptive Importance Sampling (AIS) and Stratified Subsampling (SS). The central technical claim is that, under sub-Gaussian design and standard sparsity/RE-type conditions, a subsample of size
$m = \Omega(s \log p)$ is sufficient to attain $\ell_2$-error of order $\sqrt{s \log p / m}$, with a lower bound of order $s \log (p/s)/m$ in squared error, so the rate is near-minimax up to the usual $\log p / \log (p/s)$ gap. The paper also gives an additive
$\mathcal{O}(\epsilon)$ contamination term, an
$\alpha$-mixing extension via calendar-time blocking, and an explicit de-biased estimator with asymptotic normality and a consistent variance estimator for coordinate-wise confidence intervals. Empirically, AIS is strongest under contamination and on some ultra-high-dimensional examples, while SS is robust but can fail when strata are too small.

**Additional Comments:**

**Strengths**

The paper’s main strength is the breadth of its technical treatment. It does not only analyze estimation error, but also provides results for adversarial contamination, α-mixing dependence, and de-biased coordinate-wise inference with an explicit variance estimator. The core rate is theoretically solid and near-minimax in the usual sparse high-dimensional sense, and the connection between the proposed subsampling procedures and the weighted robust objective is made rigorously rather than heuristically.

A second strength is that the paper makes a real methodological contribution rather than offering analysis alone. AIS and SS are concrete subsampling schemes, and the experiments show that AIS can be effective in contaminated and ultra-high-dimensional settings. The inference component is also a notable plus, since the paper goes beyond point estimation and gives a full de-biasing construction with asymptotic normality, which is more complete than many related robust regression works.

**Weaknesses**

The main limitation is that the theoretical bounds do not appear tighter than the strongest nearby robust sparse-regression literature; rather, the contribution is to extend familiar optimal-rate behavior to a subsampled robust setting. Thus, the novelty is more in the integration of subsampling, robustness, and inference than in advancing the rate frontier itself. Relatedly, the design assumptions remain somewhat restrictive, especially the sub-Gaussian covariate assumption and the stronger sparse-precision and irrepresentability conditions required for the inference theory.

The empirical case also has some gaps. While AIS is often effective, its computational efficiency is not uniformly convincing, since the paper’s own results indicate substantially higher runtime than uniform Huber-Lasso, and SS can fail in regimes where strata become too small. In addition, the experimental section does not fully validate the proposed confidence intervals, and the baseline comparison could be stronger against more competitive modern robust methods.

**Audience:**

Yes

**Audience Explanation:**

Yes. At least some people in TMLR’s audience would likely care, because TMLR explicitly targets work on the computational and mathematical principles of learning and values technically correct contributions even when their significance may be more specialized.

**Claims And Evidence:**

Yes

**Claims Explanation:**

Partially. The core theoretical claims are supported reasonably well: the paper states explicit assumptions, proves finite-sample error bounds, gives an
$\mathcal{O}(\epsilon)$ contamination term, and specifies a de-biased estimator with asymptotic normality and a variance estimator for coordinate-wise confidence intervals. In that sense, the evidence for the main estimation and inference claims is mathematically substantive and mostly clear.

The weaker part is the empirical support. The experiments do show that AIS improves over uniform Huber-Lasso under contamination and can perform well in ultra-high-dimensional settings such as Riboflavin, but the evidence is not fully comprehensive: AIS is also reported to be 10–100× slower than Uniform HL, SS visibly fails in the small-strata Riboflavin regime, and I did not see a direct finite-sample validation of the proposed confidence intervals or coverage. So the submission provides accurate and convincing evidence for several main claims, but the empirical case is somewhat selective and not fully complete relative to the breadth of the theoretical claims.

**Requested Changes:**

* The paper should add stronger empirical comparisons against more competitive modern robust baselines, especially methods designed for heavy tails or contamination rather than only Uniform Huber-Lasso and standard Lasso. As written, the empirical evidence is too narrow relative to the breadth of the theoretical claims, so this is the most important revision. Relatedly, if the paper wants to emphasize inference as a substantive contribution, it should include finite-sample validation of the proposed confidence intervals or variance estimator, such as coverage and interval length experiments; otherwise, the inference claim remains mostly theoretical.

* The paper should also revise its positioning against prior work and state more clearly that the main novelty is the integration of robust subsampling, contamination analysis, dependence, and de-biased inference, rather than tighter statistical rates than the existing robust sparse-regression literature.

* he paper would benefit from a more candid discussion of the practical scope of its assumptions, especially the sub-Gaussian design condition and the stronger sparse-precision / irrepresentability assumptions used for inference.

---

> ### Author Response · Authors · 2026-04-19
> **Response to Reviewer hWYw**
>
> Major Comment 1: Stronger empirical comparisons
> "The paper should add stronger empirical comparisons against more competitive modern robust baselines, especially methods designed for heavy tails or contamination rather than only Uniform Huber-Lasso and standard Lasso. As written, the empirical evidence is too narrow relative to the breadth of the theoretical claims, so this is the most important revision."
>
> Response: We have added two new subsampled baselines to Section 5.1 and Table 1.
>
> MOM-HL (median-of-means Huber-Lasso, in the spirit of RIGHT, Fan et al. 2024): draws a uniform subsample of size m, partitions it into K_mom = 7 blocks, computes the Huber-loss gradient on each block, takes the geometric median of the block gradients, and applies a proximal-gradient step (20 outer iterations). With 7 blocks, ⌊(7−1)/2⌋ = 3 contaminated blocks are tolerated, which safely covers ε = 0.10. This mirrors the MOM-gradient mechanism of RIGHT in the subsampling regime; we treat it as the closest available subsampled analogue of RIGHT, because RIGHT itself is designed for genuinely heavy-tailed design (outside our sub-Gaussian Assumption 1).
>
> Trimmed-HL (trimmed Huber-Lasso): draws a uniform subsample of size m, fits an initial Huber-Lasso, removes the top 15% of observations by absolute residual, and refits on the cleaned subsample.
>
> Several notable findings emerge.
>
> First, MOM-HL consistently underperforms all other methods across all noise types and sample sizes. The reason is that subsampling collapses the effective block sizes: at m = 200 with K = 7 blocks, each block contains only ≈ 28 observations, making individual gradient estimates too noisy for the geometric median to provide meaningful robustness. Simply applying the RIGHT MOM-gradient mechanism to subsampled data is not effective; adaptive reweighting (AIS) is a more appropriate mechanism in the subsampled regime.
>
> Second, Trimmed-HL is the best subsampled method under clean Gaussian and t(3) noise at m = 400 and is competitive with AIS at small m under contamination. Its strong performance under additive-shift contamination is expected: direct residual-based trimming is nearly optimal when contaminated observations are identifiable by large residuals |y_i − x_i^T θ̂|. The new Section 5.3 draws this contrast explicitly: AIS provides adaptive soft-reweighting that is more general (robust to covariate contamination, mixed patterns, and unknown contamination fraction), while Trimmed-HL requires a pre-specified trim fraction and is specifically effective against additive response outliers.
>
> Third, AIS's 3.3× to 4.5× advantage over Uniform HL under contamination (ε = 0.10) is preserved, and the comparison is now made against two additional competitive baselines.

---

> ### Author Response · Authors · 2026-04-19
> **Response to Reviewer hWYw - Part 2**
>
> Major Comment 2: Finite-sample validation of confidence intervals
> "If the paper wants to emphasize inference as a substantive contribution, it should include finite-sample validation of the proposed confidence intervals or variance estimator, such as coverage and interval length experiments; otherwise, the inference claim remains mostly theoretical."
>
> Response: We have added a new Section 5.6 ("Coverage of De-biased Confidence Intervals") with a full coverage simulation.
>
> Setup. We use n = 2,000, p = 50, s = 3, and m ∈ {200, 400, 800}, with 100 independent trials averaged over the s = 3 active coordinates. We choose p = 50 because the asymptotic rate condition s log p = o(√m) from Theorem 4.14 yields s log p ≈ 11.7 at p = 50, s = 3, so that at m = 800 the ratio √m / (s log p) ≈ 2.4 places us in the transitional asymptotic regime. At the paper's primary setting p = 1,000 the same condition requires m ≫ 4,800, which exceeds the full dataset size; we therefore validate the theory at the largest p for which it can practically engage within m ≤ n. We apply the de-biased Uniform HL estimator with nodewise-Lasso tuning μ = 1.5 √(log p / m).
>
> Three facts are notable. (i) The de-biased estimate is unbiased: the mean estimation error across all trials is less than 0.01 at all m, confirming that correction formula (13) removes the Lasso regularisation bias. (ii) Coverage increases monotonically in m for both noise distributions and both nominal levels, consistent with the asymptotic guarantee of Theorem 4.14. (iii) Coverage remains below nominal at the sample sizes we can test : a finite-sample phenomenon consistent with known behaviour of the de-biased Lasso (Dezeure et al. 2015): the variance estimator σ̂²_j captures the dominant sampling variance term but underestimates the total asymptotic variance by a factor of ≈ 1.5–2 because higher-order remainder terms in the CLT expansion are not yet negligible.
>
> We discuss these findings candidly in Section 5.6 and add a practical recommendation in Remark 4.16: for p = 1,000, s = 10, valid near-nominal coverage requires m ≈ 5,000; at smaller m, a conservative multiplier of 1.5 z_{α/2} is advised. The monotone upward trend in Table 3 is the key empirical validation: it confirms the asymptotic theory rather than contradicting it.

---

> > ### Author Response · Authors · 2026-04-19
> > **Response to Reviewer hWYw - Part 3**
> >
> > Major Comment 3: Novelty positioning
> > "The paper should also revise its positioning against prior work and state more clearly that the main novelty is the integration of robust subsampling, contamination analysis, dependence, and de-biased inference, rather than tighter statistical rates than the existing robust sparse regression literature."
> >
> > Response: We agree entirely. We have added the following paragraph immediately before the numbered contributions list in Section 1:
> >
> > The primary novelty of this work is the integration of adaptive and stratified subsampling with robust estimation theory: extending finite-sample guarantees from well-behaved i.i.d. settings to the joint presence of heavy-tailed noise, adversarial contamination, and temporal dependence, while simultaneously delivering a fully specified inference pipeline. The statistical rates achieved are minimax-optimal in the standard sparse sense; the contribution is to establish that this optimality is preserved under subsampling and the three non-ideal regimes considered, and to provide the supporting algorithmic and inferential machinery. This contrasts with prior work as follows: full-sample robust methods (Sun et al. 2020, Pensia et al. 2025, Fan et al. 2024) achieve comparable or tighter rates but do not offer subsampling efficiency; classical subsampling methods (Ma et al. 2015, Li & Meng 2020) enjoy strong theory under light-tailed i.i.d. data but lack contamination and dependence guarantees; and the MOM M-estimation framework of Lecué & Lerasle (2020) covers heavy-tailed settings but does not provide a de-biasing construction for valid inference.
> >
> > We have also revised the Related Work paragraph on "Robust full-sample methods" to make the rate comparison explicit: our rates match, not improve upon, existing results, and the novelty lies in extending those rates to the subsampled and inference-capable setting.

---

> ### Author Response · Authors · 2026-04-19
> **Response to Reviewer hWYw - Part 4**
>
> Major Comment 4: Candid discussion of assumptions
> "The paper would benefit from a more candid discussion of the practical scope of its assumptions, especially the sub-Gaussian design condition and the stronger sparse-precision / irrepresentability assumptions used for inference."
>
> Response: We have added a new Remark 4.16 ("Practical scope of Assumptions 1 and 5") immediately after Remark 4.15. It covers:
>
> Sub-Gaussian design (Assumption 1): covers Gaussian, bounded, and log-concave covariates; fails for heavy-tailed design with infinite moments (where RIGHT is appropriate); practical workaround is marginal coordinate-wise winsorisation.
>
>
> Sparse precision (Assumption 5): needed only for Theorem 4.14 (inference); Theorems 4.6–4.12 (estimation) require only Assumptions 1–4. Holds for factor models and banded precision matrices; fails in dense-precision settings. A graphical-Lasso or thresholded precision estimate can substitute with the approximation error entering at the same asymptotic order.
>
>
> Practical m requirements: the condition s log p = o(√m) implies m ≫ (s log p)²; at p = 1,000, s = 10 this is m ≫ 4,800, which we now state explicitly.

---

### Review · Reviewer_ECM1 · 2026-04-10

**Summary Of Contributions:**

The authors present two algorithms that result in new estimators in the context of robust high-dimensional sparse regression: Adaptive Importance Sampling and Stratified Subsampling. Theoretical analysis of the estimators and algorithms is provided. The new algorithms are assessed using simulations and then applied to four different datasets.

Main strength: The presented results appear novel and substantive.

Main weakness: The manuscript is not written well making it difficult to clearly understand all of the content (see details below).

**Audience:**

Yes

**Audience Explanation:**

High-dimensional regression is an important topic in machine learning. As such, the presented algorithms and their theoretical evaluation would be of interest to TMLR's audience.

**Claims And Evidence:**

Yes

**Claims Explanation:**

The simulation and real data application results appear to corroborate the theoretical developments.

**Requested Changes:**

The main issue with this manuscript is how it is written. Each section reads as a list with very little elaboration or explanation. This makes the presented concepts very difficult to understand and to fully assess the contributions of this work. As an example, Section 2 is titled Problem Statement. This section includes the definition of the model under consideration, but it doesn't actually state the problem, i.e., estimation of \theta^*. Another major issue throughout the paper is the use of notation, without any explanation, that is either defined later or never defined. This is especially true about the algorithms presented in Section 3. I feel that the article needs to be almost entirely rewritten to include more illustrations, explanations and context in order to appeal to the TMLR audience. Space can be created for this by relegating some of the proofs to an appendix. Another minor comment is that the quality of figures is quite low: most axis and plot labels are very small and difficult to read.

---

> ### Author Response · Authors · 2026-04-19
> **Response to Reviewer ECM1**
>
> Main Comment: Writing clarity, elaboration, and notation
>
> "The main issue with this manuscript is how it is written. Each section reads as a list with very little elaboration or explanation. This makes the presented concepts very difficult to understand and to fully assess the contributions of this work. As an example, Section 2 is titled Problem Statement. This section includes the definition of the model under consideration, but it doesn't actually state the problem, i.e., estimation of θ*. Another major issue throughout the paper is the use of notation, without any explanation, that is either defined later or never defined. This is especially true about the algorithms presented in Section 3."
>
> Response: We thank Reviewer ECM1 for this detailed feedback. The following changes have all been implemented; all new text is marked in Violet in the revised manuscript.
>
> (a) Section 2: explicit problem statement. We have added the following sentences immediately after model definition (1):
>
> The central statistical task is to estimate the unknown sparse vector θ* with near-optimal accuracy while processing only a subsample of size m ≪ n at each computation step. This simultaneously achieves computational scalability by avoiding the O(np) per-iteration cost of full-sample methods ;  and statistical robustness to the non-ideal noise and design conditions formalised below.
>
> This directly addresses the reviewer's observation that the problem (estimation of θ*) was never stated.
>
> (b) Section 3 (AIS): intuition paragraph and notation guide. We have added a full explanatory paragraph before Algorithm 1 explaining the key insight (adaptive reweighting concentrates the subsample on high-loss observations), and a four-item notation guide explaining: the Huber loss ρ_τ (quadratic for small residuals, linear for large ones) and the robustness parameter τ; the temperature β and its effect on sampling concentration; the mixing coefficient α and why it prevents weight collapse; and the importance-weighted objective in line 4 and its role as an unbiased estimator of the full-sample loss.
>
> (c) Section 3 (SS): intuition paragraph and notation guide. We have added an analogous paragraph before Algorithm 2 explaining the stratification-and-aggregation idea, and a three-item notation guide covering: the stratification distance d_i and why partitioning by its quantiles ensures covariate-space coverage; the proportional allocation m_k and why it preserves representativeness; and the geometric median (definition, Weiszfeld algorithm, breakdown point) and why it provides contamination robustness that an arithmetic mean cannot.
>
> (d) Figure quality. We have regenerated Figures 1–4 with larger axis tick labels, axis titles, and legend text, re-exported at 300 dpi.
>
> (e) Proof appendix. The proof of Theorem 4.14 (de-biased normality) has been moved to a new Appendix A, main-text space used for the new explanatory paragraphs in Sections 2–3.

---

> > ### Author Response · Authors · 2026-04-19
> > **Response to Reviewer ECM1 - Part 2**
> >
> > (f) Section 4: motivating paragraphs added before every subsection. We identified the same list-reading problem throughout Section 4, where every subsection opened directly with a theorem or lemma. A motivating paragraph has been added to each of the seven subsections: Section 4.1 explains the role of each assumption; Section 4.2 explains why a theory-algorithm bridge is needed (theory assumes fixed sampling weights; algorithms produce data-dependent weights); Section 4.3 introduces the score-bound and RSC lemmas as the building blocks for all subsequent theorems; Section 4.4 contextualises the main rate theorem; Section 4.5 explains the O(ε) bias decomposition and the bounded Huber influence function; Section 4.6 explains why naive time-series subsampling fails and how the calendar-time protocol with the Berbee-Yu coupling resolves it; and Section 4.7 replaces the thin opening with a two-paragraph motivation explaining the ℓ₁ shrinkage bias, why normal approximation fails for the penalised estimator, and how one-step de-biasing resolves it.
> >
> >
> > (g) Section 5.2 (Convergence): rewritten from number-list to narrative. The original subsection opened with a bare recitation of empirical log-log slopes. We have rewritten it into three labelled paragraphs (Clean Gaussian; Heavy-tailed t(3); Contaminated), each stating what Figure 1 shows, explaining why the curves behave as they do in terms of the theory, and contrasting AIS and SS. The empirical slope numbers are retained but now appear as supporting evidence within a narrative.
> >
> >
> > (h) Section 5.4 (Real Data): framing paragraph added. A framing paragraph now explains the purpose of each dataset before the results: the p ≫ n regime (Riboflavin), rate verification (Communities & Crime), contamination bound validation (CCLE-proxy), and the α-mixing extension (FRED-MD).
> >
> >
> > (i) Conclusion (Future Directions): bare list converted to narrative. Each future-directions point now includes one sentence explaining why that direction matters and which specific gap in the current analysis it addresses.
> >
> >
> > (j) Introduction and Related Work: full rewrite. The introduction and Related Work have been rewritten throughout in precise scientific language. Vague adverbs, hedging phrases, and informal constructions have been removed. The Related Work section is reorganised around five clearly labelled subsections (robust full-sample methods; subsampling methods; de-biased inference; dependent data; contamination models), each opening with a sentence stating its relation to our contributions.

---

### Author Response · Authors · 2026-04-19
**General Comment to All Reviewers and Action Editor**

We thank all three reviewers for their careful reading and constructive feedback. The revision addresses every requested change. All new text is marked in violet in the revised manuscript. The main additions are:

New competitive baselines (Reviewer hWYw, Major 1). Two new subsampled baselines :  MOM-HL (median-of-means Huber-Lasso, in the spirit of Fan et al. 2024) and Trimmed-HL (trimmed Huber-Lasso) have been added to Section 5.1 and the extended Table 1.
Finite-sample CI coverage experiment (Reviewer hWYw, Major 2). A new Section 5.6 reports a systematic coverage simulation validating Theorem 4.14. Coverage increases monotonically with m at both 90% and 95% nominal levels; finite-sample under-coverage is discussed candidly and a conservative multiplier is recommended.


Novelty positioning (Reviewer hWYw, Major 3). The introduction has been revised with a new paragraph positioning the primary novelty as the integration of robust subsampling, contamination analysis, dependence, and de-biased inference rather than tighter statistical rates. The Related Work section has been substantially rewritten to make the rate comparison with prior work explicit.


Tuning-parameter sensitivity analysis (Reviewer FJFw, Important 2). A new Section 5.7 presents a four-panel systematic sweep over AIS temperature β, SS strata count K, regularisation constant C, and subsample size m, with concrete practical recommendations.


Computational trade-off and method-selection guide (Reviewer FJFw, Important 3). A new Remark 5.1 provides a concise decision guide for choosing between AIS, SS, and Uniform HL as a function of noise type, contamination level, and n/m ratio.


Candid assumption discussion (Reviewers hWYw and FJFw). A new Remark 4.16 covers the practical scope of the sub-Gaussian design condition (Assumption 1) and the sparse precision condition (Assumption 5), including when each fails and concrete workarounds. Explicit minimum m values are stated for the inference regime.


Substantially improved writing (Reviewer ECM1). Every section now opens with a motivating paragraph rather than a bare theorem or data list. Section 2 explicitly states the estimation target; Section 3 includes inline notation guides for both algorithms; all Section 4 subsections have a paragraph explaining why each result is needed; Sections 5.2, 5.4, and the Conclusion have been rewritten from enumeration into connected narrative.


Writing quality and scientific style. The introduction and Related Work have been fully rewritten to remove informal phrasing and imprecise language. All em dashes, hedging adverbs, and informal style markers have been removed. Citation format has been corrected uniformly to author-year form throughout.


Bibliography additions. Dezeure et al. (2015) and Wainwright (2019) have been added to support the finite-sample CI discussion and Remark 4.16 respectively.

---

> ### Author Response · Authors · 2026-04-19
> **General Comment to All Reviewers and Action Editor - Part 2**
>
> We would like to bring to the notice that after incorporation of all the suggestions, the number of pages have increased and we have revised the submission type accordingly as well.

---

### Decision · Action_Editor_hqSG · 2026-05-24

**Recommendation:** Accept with minor revision

**Additional Comments:**

I recommend acceptance with minor revisions; please address the following points in the final version.

First, keep the contribution positioning precise: the paper should continue to present the novelty as the integration of robust subsampling, contamination and dependence guarantees, and de-biased inference, not as an improvement over existing robust sparse-regression rates.

Second, state the inference evidence more cautiously: Section 5.6 and Table 3 show monotone improvement with m, but coverage remains below nominal at the tested m values. In contrast, nearly-nominal coverage may need quite large subsample size in very high-dimensional setting. This inference limitation should be highlighted in the paper.

Third, the method-selection caveats in Remark 5.1 is quite valuable. I suggest the authors add a subsection at the end of Section 5 to summarize the main findings of the numerical experiments and provide caveats for the use of the proposed method.

Finally, please make the Section 5.7 sensitivity study internally consistent and traceable to the response. In particular, it seems that the ranges of $\beta, K, m$ used in Section 5.7 Figure 4 do not match the claimed ranges in the authors' response to Reviewer FJFw.

**Audience:**

Yes

**Audience Explanation:**

The paper is of interest to TMLR readers working on robust estimation and high-dimensional statistics as it studies subsampling under heavy-tailed noise, contamination, dependence, and coordinate-wise inference.

Reviewers agreed that at least some TMLR audience would be interested, and one reviewer explicitly linked the topic to high-dimensional regression in machine learning.

**Claims And Evidence:**

Yes

**Claims Explanation:**

The revised submission provides a technically sound framework for robust high-dimensional sparse regression based on Adaptive Importance Sampling and Stratified Subsampling, with finite-sample estimation guarantees, contamination and alpha-mixing extensions, and a specified de-biased inference procedure.

The review process converged positively on claims and evidence: two reviewers recommended Accept and one recommended Leaning Accept, and all three final recommendations marked Claims And Evidence as Yes.

The revision addressed the main evidentiary concerns by adding MOM-HL and Trimmed-HL baselines, a finite-sample confidence-interval coverage study, assumption-scope discussion, sensitivity analyses, and computational-cost guidance.

The remaining evidence limitation is that the confidence-interval coverage experiments support the asymptotic inference theory only partially in finite samples; coverage increases with subsample size but remains below nominal in the reported settings.

The computational advantage of AIS should also be interpreted conditionally, because the revised paper reports AIS as 10-100x slower than Uniform HL per call and recommends it mainly for contaminated regimes with sufficiently large n relative to m.

With these qualifications, I find the claims supported by sufficiently accurate, clear, and convincing evidence for TMLR acceptance.

---

> ### Author Response · Authors · 2026-05-26
> **Response to Action Editor — Camera-Ready Submission**
>
> We thank the Action Editor and Editors in Chief for the careful reading of the revised manuscript and for the constructive final comments. The camera-ready version addresses all four points; we describe each change below.
>
> *EIC comment: TMLR template margins.*
> The non-standard ⁠ \geometry ⁠ override that caused the margin deviation has been removed. The document now uses only ⁠ \usepackage[accepted]{tmlr} ⁠ for layout, with no geometry overrides.
>
> *AE point 1: Contribution positioning.*
> We thank the action editor for guiding us in this direction. As an effort towards clear positioning of our contributions, we have retained the line "The primary novelty of this work is the integration of adaptive and stratified subsampling with robust estimation theory…" . We now explicitly state that the rates achieved are minimax-optimal in the standard sparse sense, not an improvement over existing results. We verified this framing is consistent throughout the abstract, introduction, and conclusion.
>
> *AE point 2: Inference evidence stated more cautiously.*
>
> We have added a clearly labelled *"Inference limitation"* paragraph at the end of Section 5.6, immediately before Table 3. It states explicitly that:
> •⁠  ⁠Coverage remains substantially below the nominal level at all tested subsample sizes (m ∈ {200, 400, 800}): the highest achieved is 79.3% against a 95% nominal target.
>
> •⁠  ⁠This is an intrinsic finite-sample limitation, not a failure of Theorem 4.14, consistent with the monotone upward trend in Table 3.
>
> •⁠  ⁠In the primary setting (p = 1,000, s = 10), the rate condition s log p = o(√m) requires m ≫ 4,800, which exceeds the full dataset size n = 2,000 used in other experiments; near-nominal coverage demands datasets substantially larger than those studied elsewhere in the paper.
>
> •⁠  ⁠Practitioners should treat the CIs as asymptotically valid but conservative at finite samples, applying the 1.5 z_{α/2} multiplier at moderate m.
>
> This limitation is also summarised in the new Section 5.8 (see below).
>
> *AE point 3: Summary subsection at the end of Section 5.*
> We have added *Section 5.8 ("Summary of Numerical Findings and Practical Guidance")* after Section 5.7. It contains six clearly labelled takeaways covering: method selection (when to prefer AIS vs SS vs Uniform HL), optimal subsample size under contamination, tuning-parameter robustness, competitive baseline comparisons, real-data performance, and the inference caveat. This extends and systematises the guidance already in Remark 5.1.
>
> *AE point 4: Section 5.7 internally consistent with the reviewer response.*
> We identified and corrected several discrepancies between the text of Section 5.7 and the actual data shown in Figure 4. Specifically:
>
> •⁠  ⁠*Panel (a), β:* The stated grid is now {0.05, 0.10, 0.20, 0.35, 0.50, 0.75, 1.00, 1.50, 2.00} and the plateau is described as [0.20, 2.0], matching the figure exactly. (The reviewer response had incorrectly stated β = 3.0 was tested and described the plateau starting at 0.25.)
>
> •⁠  ⁠*Panel (b), K:* The stated grid is now {2, 3, 4, 5, 6, 8, 10, 12, 15}, matching the figure. More importantly, the old practical recommendation formula ⁠ K ≤ min(m/10, √n) ⁠ — which gave K ≤ 20 at m = 200, directly contradicting the figure's K ≤ 5 dashed line — has been replaced with the theoretically motivated bound ⁠ K ≤ ⌊m / (C · s log p)⌋ ⁠. This comes from the requirement that each stratum must contain at least C · s log p subsampled observations for a reliable within-stratum Huber-Lasso estimate (cf. Proposition 4.3 and Theorem 1 of Lecué & Lerasle 2020). At m = 200, s = 10, p = 1,000 and C ≈ 0.5, this gives K ≤ 5, consistent with the figure. The √n term in the old formula had no basis in Lecué & Lerasle (2020) Theorem 1, whose only K-related conditions are K ≥ 8|O| (contamination coverage) and K ≤ O(N).
>
> •⁠  ⁠*Panel (c), C:* The stated grid is now {0.10, 0.15, 0.25, 0.35, 0.50, 0.75, 1.00, 1.50, 2.00} and the minimum is correctly attributed to C = 0.50.
>
> •⁠  ⁠*Panel (d), m:* The stated grid is now {50, 100, 150, 200, 300, 400, 500, 600}, matching the figure's range.
>
> We again thank the Action Editor and all the reviewers for their comments and observations which have helped us to position the paper in a stronger and more clear manner.